



# The Information Content of Dense Carbon Dioxide Measurements from Space: A High-Resolution Inversion Approach with Synthetic Data from the OCO-3 Instrument

Dustin Roten[1], John C. Lin[1], Lewis Kunik[1], Derek Mallia[1], Dien Wu[2], Tomohiro Oda[3,4,5], and Eric A. Kort[6]

[1]Department of Atmospheric Sciences, University of Utah, Salt Lake City, UT
[2]Division of Geological and Planetary Sciences, California Institute of Technology, Pasadena, CA
[3]The Earth from Space Institute, Universities Space Research Association, Columbia, MD, USA
[4]Department of Atmospheric and Oceanic Science, University of Maryland, College Park, MD, USA
[5]Graduate School of Engineering, Osaka University, Suita, Osaka, Japan
[6]Climate and Space Sciences and Engineering, University of Michigan, Ann Arbor, MI, USA
**Correspondence:** Dustin Roten (dustin.roten@utah.edu)

**Abstract.** Bottom-up accounting methods of carbon dioxide ($CO_2$) emissions can provide high-resolution emissions estimates at a global scale; however, the necessary in situ observations to verify these emissions are limited in coverage. Space-based observations of $CO_2$ in the Earth's atmosphere expand this coverage to a near-global scale to inform carbon cycle science and record emission trends. This work applied an observing system simulation experiment (OSSE) to characterize the flux

information contained in "Snapshot Area Map" (SAM) $CO_2$ measurements from the Orbiting Carbon Observatory-3 (OCO-3). Unlike previous space-based carbon-observing systems, OCO-3 SAMs provide spatially dense observations of $CO_2$ over targeted urban areas at unprecedented coverage. A Bayesian inversion using synthetic data was applied to these SAMs to explore their effectiveness in optimizing estimates of fossil fuel $CO_2$ ($FFCO_2$) emissions from the Los Angeles Basin. Results demonstrated that errors in the locations of large point sources diminished the inversion's ability to reduce errors at the sub-

city-level. Furthermore, reductions in atmospheric transport error exacerbated these issues. Only after geolocation errors in large point source locations were removed and atmospheric transport error was reduced did individual SAM observations provide modest corrections to prior flux estimates. The aggregation of multiple SAMs proved to be effective in reducing systematic errors in manufacturing- and transportation-related estimates, demonstrating the need for similar measurements in future space-based missions.

# 1 Introduction

Atmospheric carbon dioxide ($CO_2$) is a key driver of global warming (Zhong and Haigh, 2013), and human activity is a significant source of this greenhouse gas (GHG) through the combustion of fossil fuels (Masson-Delmotte et al., 2021). Atmospheric $CO_2$ has increased notably in the past decades (Keeling and Keeling, 2017), driving policy-based efforts to reduce carbon emissions from anthropogenic sources. A notable example of such international policy is the United Nation's Paris Agreement



(2015) aimed at keeping global warming below 1.5°C (Masson-Delmotte et al., 2018). At the city-level, the C40 Cities Climate Leadership Group brings together local leaders from around the globe to focus on climate issues (Davidson et al., 2019). The governing bodies, who are signatories of these agreements, commit to the use of carbon accounting strategies and emissions reduction goals, but the ability to monitor and verify downward trends in $CO_2$ at high spatiotemporal resolution while attributing them to societal changes remains challenging (Janssens-Maenhout et al., 2020).

Anthropogenic carbon accounting uses "bottom-up" and "top-down" methods to constrain global $CO_2$ fluxes despite existing limitations. A typical bottom-up approach uses statistical methods and energy use information to construct spatially explicit inventories. Typically, multiple data streams from various economic sectors are used to synthesize estimates of total emissions which are then spatially distributed across geographic regions. Several bottom-up products exist for characterizing the distribution of anthropogenic $CO_2$ emissions at local, national, and global scales. Examples include: Hestia [local; Gurney et al. 30 (2019)], Vulcan 3.0 [national; Gurney et al. (2020a)], the Emissions Database for Global Atmospheric Research (EDGAR), and the Open-source Data Inventory for Anthropogenic $CO_2$ (ODIAC) [global; Janssens-Maenhout et al. (2019); Oda et al. (2018)]. These inventories serve as essential inputs for model simulations and references for climate reduction policies; however, their construction methods vary, as many rely on self-reported information and physical proxies to allocate spatial distributions of emissions. For this reason, considerable differences exist between inventories (Hutchins et al., 2016; Gately and Hutyra, 2017; 35 Oda et al., 2018, 2019). Therefore, these products are often compared with atmospheric observations.

Top-down constraints on $CO_2$ fluxes use atmospheric observations to infer source emission rates. Ground-based networks provide high-frequency $CO_2$ observations, although the density of network sites is often sparse. Examples of such networks include the Utah Urban $CO_2$ Network [UUCON; Bares et al. (2019)], the Indianapolis Flux Experiment [INFLUX; Davis et al. (2017)], the Berkeley Atmospheric $CO_2$ Observation Network [BEACO$_2$N; Shusterman et al. (2018)] and the Munich 40 Urban Carbon Column network [MUCCnet; Dietrich et al. (2021)]. All are aimed at constraining urban $CO_2$ emissions at city-level and sub-city-level scales (Mitchell et al., 2018b). Novel approaches to increase the spatial coverage of these observations include mobile platforms such as light rail (Mitchell et al., 2018a) and on-road mobile measurements (Lee et al., 2017). Atmospheric transport mixes $CO_2$ from a variety of sources, producing a superimposed downwind mixing ratio at the receptors of these measurement networks. Through inverse modeling methods, information about the sources of $CO_2$ signals can be 45 retrieved from observations, revealing insights into the dynamics of these signals and the assumptions made in the construction of prior estimates (Rodgers, 2000). Previous studies have used long-running time series of surface-based $CO_2$ measurements in their calculations to optimize prior emissions estimates from inventories at the local/regional level (Lauvaux et al., 2016; Kunik et al., 2019; Mallia et al., 2020; Turner et al., 2020; Lauvaux et al., 2020), but surface observations are limited to a handful of cities.

In an effort to increase the spatial coverage of atmospheric $CO_2$ observations and work towards global monitoring of sources and sinks, space-based platforms have come online in recent years. Current missions include NASA's Orbiting Carbon Observatory [OCO; Crisp et al. (2004)], JAXA's Greenhouse Gases Observing Satellite [GOSAT; Yokota et al. (2009)], and CNES's MicroCarb (Bertaux et al., 2020). Additionally, several instruments are in planning stages with notable examples being NASA's GeoCARB (Polonsky et al., 2014) and ESA's $CO_2$M (Sierk et al., 2019) which are tentatively scheduled for launch in 2024





and 2025, respectively. Although the original intention of space-based observatories was to monitor columned-averaged $CO_2$ ($XCO_2$) on a global scale, these instruments are also effective at constraining anthropogenic emissions at regional and local scales. In particular, observations from OCO-2 were used to confirm relationships between city density and $CO_2$ emissions that were previously demonstrated using only emissions inventories (Wu et al., 2020; Yang et al., 2020) while another study identified an underestimation of $CO_2$ flux along the Nile River Delta (Shekhar et al., 2020). These studies constrained emissions

from a whole-city perspective, not addressing sub-city-level contributions or individual sectors. Bayesian synthesis inversions have also been applied to $XCO_2$ observations at city-level (Ye et al., 2020; Lei et al., 2021) and global (Wang et al., 2018; Lespinas et al., 2020) scales to optimize prior surface flux estimates (inventories) based on instrument observations.

  Some of the space-based platforms listed above provide narrow transects of observations in a north-south orientation while others observe wider tracks at lower resolutions. Typical target revisit times range from three to 16 days (Kataoka et al.,

2017); however, this is not the case for the latest installment of the OCO mission: OCO-3. This instrument was launched in 2019 and implements new machinery to increase transect widths. OCO-3 is housed on the International Space Station and is capable of accumulating spatially dense $XCO_2$ observations over target locations in a matter of minutes, creating "Snapshot Area Maps" (SAMs) (Eldering et al., 2019; Taylor et al., 2020). While OCO-2 can provide a single 10 km wide track of $\sim 2$ km-spaced soundings over an area of interest every 16 days at $\sim$13:30 local time, OCO-3's shorter revisit time provides

multiple 80 km $\times$ 80 km SAMs over the same area at varying times of day and similar resolution, observing three times the surface flux signal as OCO-2 (Kiel et al., 2021). The increased spatiotemporal coverage from SAMs has the potential to provide unprecedented insights into the emission dynamics of many megacities around the world, likewise increasing the information available for inversion schemes.

  To investigate the utility of the information content provided by OCO-3 SAMs, an observing system simulation experiment

(OSSE) was designed and implemented over the Los Angeles Basin (referred to hereafter as LA). The use of synthetic data from OCO-3 eliminated the potential for systematic biases from local $CO_2$ emissions reductions during the COVID-19 pandemic lockdowns (Weir et al., 2021) and biases in preliminary data from OCO-3 (Buchwitz et al., 2021; Taylor et al., 2020). The use of LA as a test bed ensured the availability of multiple high-resolution emission inventories and meteorological data. Furthermore, LA's large urban population, represented with emission inventories, provided sizable anthropogenic enhancements in $XCO_2$-

space. Here, a previously-established Bayesian inversion OSSE scheme (Kunik et al., 2019) was applied to synthetic SAMs to investigate the effects of large localized uncertainty, spatial resolution of input information, atmospheric transport error, and systematic bias on their information content. Results demonstrated the potential for SAMs to inform prior flux estimates at the sub-city-level with implications for the urban carbon cycle, construction of emissions inventories, and locally targeted emissions reduction policies. As the constellation of $CO_2$-observing satellites continues to grow, this work emphasizes the

importance of spatially dense $XCO_2$ retrievals over megacities around the world. Although a few cities have operational ground-based observing systems, these space-based instruments provide a means to optimize bottom-up estimates and verify current/future emissions agreements at a variety of political scales (Pan et al., 2021).



## 2   Methods

### 2.1   Defining the Flux Domain

This work applied an OSSE over LA, an area of varied topography with the Pacific Ocean to the southwest. The U.S. Census Bureau's *Annual Estimates of the Residential Population* estimates the 2021 population of this area to be 18.8 million. To incorporate the size of larger SAMs, the flux domain encompassed the entirety of the LA Basin, San Fernando, and San Bernardino valleys. Depictions of this area and the domain are presented in **Fig. 1 (top)**.

### 2.2   Emission Inventories

The Bayesian inversion scheme used in this OSSE optimized a user-supplied *prior* flux estimate to better approximate "*true*" fluxes within LA. Under this scheme, the "*true*" fluxes are explicitly used to derive synthetic observations of fossil fuel based $CO_2$ (FFCO$_2$) signals, making them "known" by the user. Here, two spatially explicit $CO_2$ emissons inventories were used: Vulcan 3.0 and ODIAC-VIIRS. Vulcan 3.0 synthesizes data from multiple sources to represent sector-specific information at an hourly scale and (roughly) 1 km × 1 km resolution (Gurney et al., 2020b, a) whereas ODIAC-VIIRS uses global nighttime
light observations from space as a proxy for the distribution of non-point source emissions at the same resolution (Oda et al., 2021). FFCO$_2$ ("*true*") fluxes from LA were represented using nine sectors from the latest available year (2015) of Vulcan 3.0 but, for the purposes of this study, commercial maritime and coastal emissions have been omitted since off-shore emissions are not included in ODIAC-VIIRS. For simplicity, the remaining sectors were aggregated into four categories: "Power Industry", "Manufacturing", "Transportation", and "Buildings". Specifics of this aggregation are included in **Tab. 1**.

**Table 1.** Similar Vulcan 3.0 emissions sectors are aggregated into four categories. Category names are in the left column and the corresponding Vulcan sectors are in the right column.

| Category | Vulcan 3.0 Sectors |
|---|---|
| Power Industry | Electricity Production |
| Manufacturing | Industrial, Cement Production |
| Transportation | Onroad, Non-road (ATV), Rail, Airport (Taxi, Takeoff) |
| Buildings | Residential, Commerical |

The version of ODIAC used as the *prior* estimate was supplemented by the Visible Infrared Imaging Radiometer Suite (VIIRS) onboard the Suomi National Polar-Orbiting Partnership (NPP) operated by the United States' National Oceanic and Atmospheric Administraion (NOAA) (Oda et al., 2021). This particular iteration of ODIAC provided annual surface flux estimates of FFCO$_2$ at roughly 1 km × 1km resolution. Since this inventory uses global nighttime light observations to distribute emission estimates, sector-specific information was not available. To obtain the same temporal resolution as Vulcan
3.0, the annual ODIAC-VIIRS data was first downscaled to monthly timesteps. This initial downscaling was accomplished by creating spatial weighting factors via ratios of monthly and annual data from an alternative version of ODIAC for year 2019

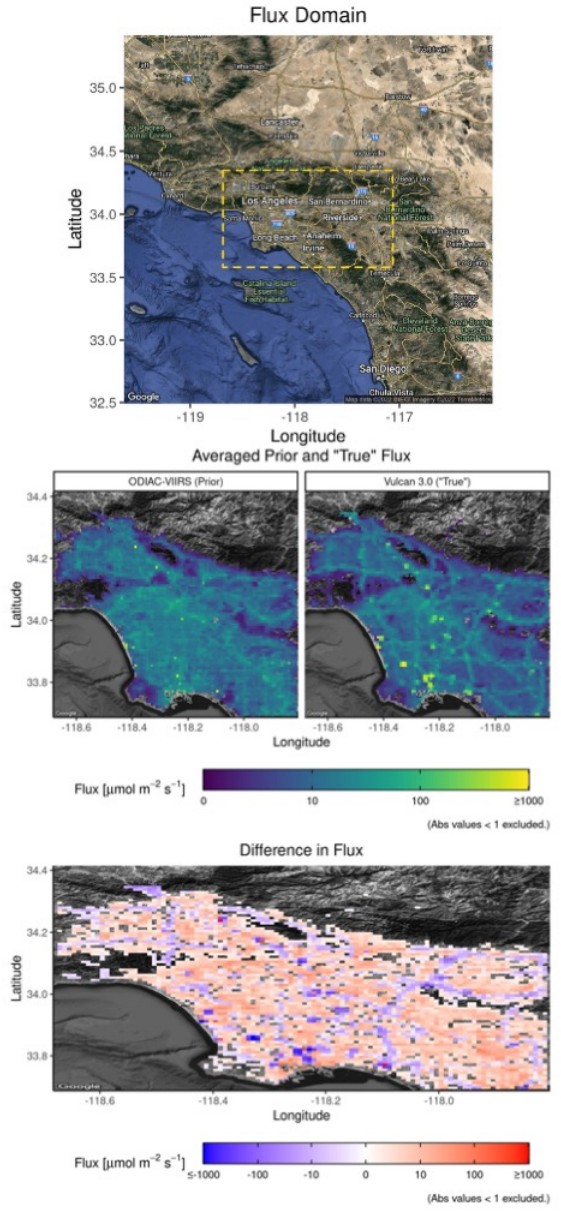

**Figure 1.** [Top] The urban domain of interest (yellow, dashed box) located in the Los Angeles Basin in southern California, USA. The domain also encompasses the San Fernando and San Bernardino Valleys. [Middle] Mean flux distributions of two $CO_2$ emission inventories for the Los Angeles Basin are presented here. On the left, the mean flux of the ODIAC-VIIRS inventory is displayed. These emissions are from year 2019. On the right, surface flux as reported by the Vulcan 3.0 emission inventory is displayed. This is from year 2015. [Bottom] Displayed here are the differences between the ODIAC-VIIRS and Vulcan 3.0 emission inventories (ODIAC-VIIRS minus Vulcan 3.0). (Map Data © 2022 INEGI Imagery ©2022 TerraMetrics)





[ODIAC2020b; (Oda, 2015)]. Weekly and hourly temporal downscaling was achieved by applying the Temporal Improvements for Modeling Emissions by Scaling (TIMES) scheme developed by Nassar et al. (2013). Averaged distributions for both inventories are presented along with their spatial differences in **Fig. 1 (middle, bottom)**. The difference plot highlights the higher area emissions present in ODIAC, the well-defined interstate highways in Vulcan 3.0, and Vulcan's large point source (LPS) emissions along the southern coast associated with industrial sources.

### 2.3 Generating Pseudo-Enhancements with X-STILT

Representing the "*true*" emissions with Vulcan 3.0 allowed for the investigation of sector-specific influences in $XCO_2$-space since the relative contribution of each sector to the "observed" $CO_2$ signal is explicitly known in this scheme. First, preliminary SAMs were obtained from the OCO-3 instrument (Taylor et al., 2020; Gunson and Eldering, 2020). The geolocations of their soundings were then used as the locations of the pseudo-observations while their associated $XCO_2$ values were generated with the column-averaged version of the Stochastic Time Inverted Lagrangian Transport (X-STILT) model (Wu et al., 2018). This time-inverted model distributes particles vertically within an atmospheric column to represent air parcels. Using supplied meteorological fields, these particles are propagated backwards in time where their locations and duration below the atmospheric boundary layer contribute to an influence footprint for the initial column location. The influence footprints generated for each OCO-3 sounding were then convolved with the Vulcan 3.0 emission inventory to determine the "*true*" emissions contributions to each column observation over LA. Details of the footprint construction and convolution process are provided by Wu et al. (2018) and are briefly described as:

$$f_w(\mathbf{x}_{n,r}, t_{n,r}|x_i, y_j, t_k) = \frac{m_{\mathrm{air}}}{h\overline{\rho}(x_i, y_j, t_k)} \frac{1}{N_{\mathrm{tot}}} \sum_{p=1}^{N_{\mathrm{tot}}} \Delta t_{p,i,j,z \leq h} \times PW(n,r) \times AK_{\mathrm{norm}}(n,r). \tag{1}$$

Here, a column-weighted footprint (denoted as $f_w$), combined across $n$ vertical levels, for a column location ($\mathbf{x}_{n,r}$), and time ($t_{n,r}$) is sensitive to surface flux at location $x_i, y_j$ and time $t_k$. Determining the spatial magnitudes of this sensitivity depends on the molar mass of dry air ($m_{\mathrm{air}}$), the height of the mixing layer ($h$), and the average density of the air below $h$ at a particular location and time, $\overline{\rho}(x_i, y_j, t_k)$. The time spent by each particle ($\Delta t_p$), whose altitude is below the mixing layer ($z \leq h$), over a grid position ($i, j$) contributes to the sensitivity footprint during backwards propagation. This influence is averaged across the total number of particles released ($N_{\mathrm{tot}}$). These calculations are further scaled by a pressure weighting function $PW(n,r)$ and averaging kernel $AK_{\mathrm{norm}}(n,r)$ which are specific to the OCO-3 instrument and reported in the vEarly data (Gunson and Eldering, 2020; Taylor et al., 2020).

The influence footprints generated from this process were convolved with emission inventories such that:

$$\Delta XCO_2 = \sum_{i,j,k} \Delta XCO_2(\mathbf{x}_{n,r}, t_{n,r}|x_i, y_j, t_k) = \sum_{i,j,k} f_w(\mathbf{x}_{n,r}, t_{n,r}|x_i, y_j, t_k) \cdot F(x_i, y_j, t_k). \tag{2}$$

The enhancement ($\Delta XCO_2$) at a particular column location and time ($\mathbf{x}_{n,r}, t_{n,r}$) is determined by the multiplication of cell-wise surface fluxes ($F(x_i, y_j, t_k)$) and footprint values. The summation of these elements provides the column-averaged influ-





ence (ppm). Since SAMs are typically collected over a span $< 10 \, \mathrm{min}$, the timestamps of their soundings were averaged to give each SAM a unique observation time. This simplified the inversion process by using the same $t_{n,r}$ values across all soundings and synchronized $t_k$ values across all fluxes.

The X-STILT model was driven by $3 \, \mathrm{km}$ resolution High-Resolution Rapid Refresh (HRRR) data (Benjamin et al., 2016; Rolph et al., 2017) and a $6 \, \mathrm{km}$ column height was used, as characterized in Wu et al. (2018). An influence footprint was generated for each sounding by distributing co-located columns of particles backwards in time for $18 \, \mathrm{h}$ while averaging at 1 h intervals. In most cases, particles propagated out of the LA Basin at later timesteps. This is demonstrated in **Fig. 2** where selected *backwards* timesteps were averaged across all X-STILT footprints used in this work. The plots from later timesteps

show particles following northerly winds backwards in time. To create urban pseudo-enhancements, these footprints were convolved with the Vulcan 3.0 time series and the addition of random instrument error, $\varepsilon \sim \mathbf{N}(0, 0.23 \, \mathrm{ppm})$ was added to each signal. This error was determined by Kiel et al. (2021), who compared OCO-3 observations with atmospheric column measurements of $XCO_2$ from a Total Carbon Column Observing Network [TCCON, Wunch et al. (2011)] site in LA. Between the available OCO-3 and HRRR data, 11 SAMs were generated ranging from February to October of 2020 (see **Fig. 3**). Large

enhancement values of $\geq 4 \, \mathrm{ppm}$ are localized along the southern coast and are indicative of LPSs in the area. Further inland, mid-range values of roughly 2-3 ppm are widely distributed across the basin. Moving to the northwest, enhancements approach 0 ppm, as the associated X-STILT footprints have little to no interaction with the emissions domain.

### 2.4    The Bayesian Inversion Scheme

The application of Bayesian statistics in atmospheric inversions is a well-established method of extracting information from

surface- and space-based $CO_2$ observations (Rodgers, 2000). By representing *a priori* emission values, observed atmospheric concentrations, influence footprints, and associated uncertainties in a cost function, $L_s$, an optimized *posterior* estimate is produced. Tarantola (2005) defines this cost function such that:

$$L_s = \frac{1}{2}(\boldsymbol{z} - \mathbf{H}\hat{s})^{\mathrm{T}}\mathbf{R}^{-1}(\boldsymbol{z} - \mathbf{H}\hat{s}) + \frac{1}{2}(\hat{s} - \boldsymbol{s}_p)^{\mathrm{T}}\mathbf{Q}^{-1}(\hat{s} - \boldsymbol{s}_p). \tag{3}$$

The first term addresses estimates and error in $XCO_2$-space. Here, $\boldsymbol{z}$ represents a vector of $\Delta XCO_2$ from a given SAM (or

SAMs) with its length equal to the number of available soundings. Footprints from X-STILT are represented by the matrix $\mathbf{H}$ such that each row contains the vectorized footprint information from $f_w(\mathbf{x}_{n,r}, t_{n,r} | x_i, y_j, t_k)$. Thus, the dimensions of $\mathbf{H}$ are dictated by the number of available footprints (rows, equivalent to the length of $\boldsymbol{z}$) and number of cells in the domain of interest across all timesteps (columns). The *posterior* estimation of the surface flux is represented by the vector $\hat{s}$ where its length is also equal to the number of cells multiplied by the number of backwards timesteps in the domain of interest. Finally, $\mathbf{R}$ reflects

uncertainties in $\Delta XCO_2$ observations from various components. These errors are included along the diagonal of $\mathbf{R}$ with each element corresponding to an observation. Specific errors included in the $\mathbf{R}$ matrix and the construction of off-diagonal elements are discussed in **Sect. 2.7**.



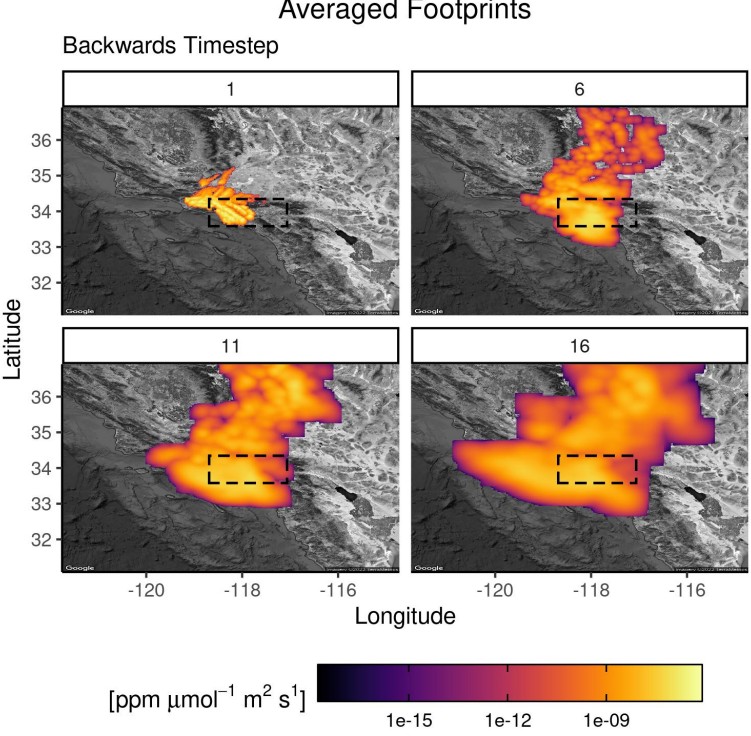

**Figure 2.** Here, all footprints are averaged at the timestep level with each panel representing the number of hours prior to the SAM observation. The emissions domain is indicated by the black dashed rectangle.

The second term on the right hand side of **Eqn. 3** addresses uncertainties in surface flux as part of the cost function, $L_s$. Here, the *prior* estimate of the surface emissions from the domain is represented by $\boldsymbol{s}_p$ which contains $F(x_i, y_j, t_k)$ in vector form. Flux uncertainty is represented by the matrix $\mathbf{Q}$ which has rows/columns equal to the length of $\boldsymbol{s}_p$. The construction of $\mathbf{Q}$ is further discussed in **Sect. 2.7**. The optimized *posterior*, $\hat{s}$, will minimize the cost function (**Eqn. 3**) and can be determined analytically:

$$\hat{s} = \boldsymbol{s}_p + (\mathbf{HQ})^{\mathrm{T}}(\mathbf{HQH}^{\mathrm{T}} + \mathbf{R})^{-1}(\boldsymbol{z} - \mathbf{H}\boldsymbol{s}_p). \tag{4}$$

Using the formulation of the pseudo-observations such that $\boldsymbol{z} = \mathbf{H}\boldsymbol{s}_t + \varepsilon$, **Eqn. 4** can be written so the "*true*" emissions ($s_t$) and instrument error are included:

$$\hat{s} = \boldsymbol{s}_p + (\mathbf{HQ})^{\mathrm{T}}(\mathbf{HQH}^{\mathrm{T}} + \mathbf{R})^{-1}[\mathbf{H}(\boldsymbol{s}_t - \boldsymbol{s}_p) + \varepsilon]. \tag{5}$$

Systematic bias correction provided by cumulative observations can be quantified by making a substitution such that $\mathbf{K} = \mathbf{H}\boldsymbol{s}'_p$. The vector $\boldsymbol{s}_p$ is replaced with a matrix of equal row length but surface fluxes are disaggregated by sector, each making up a



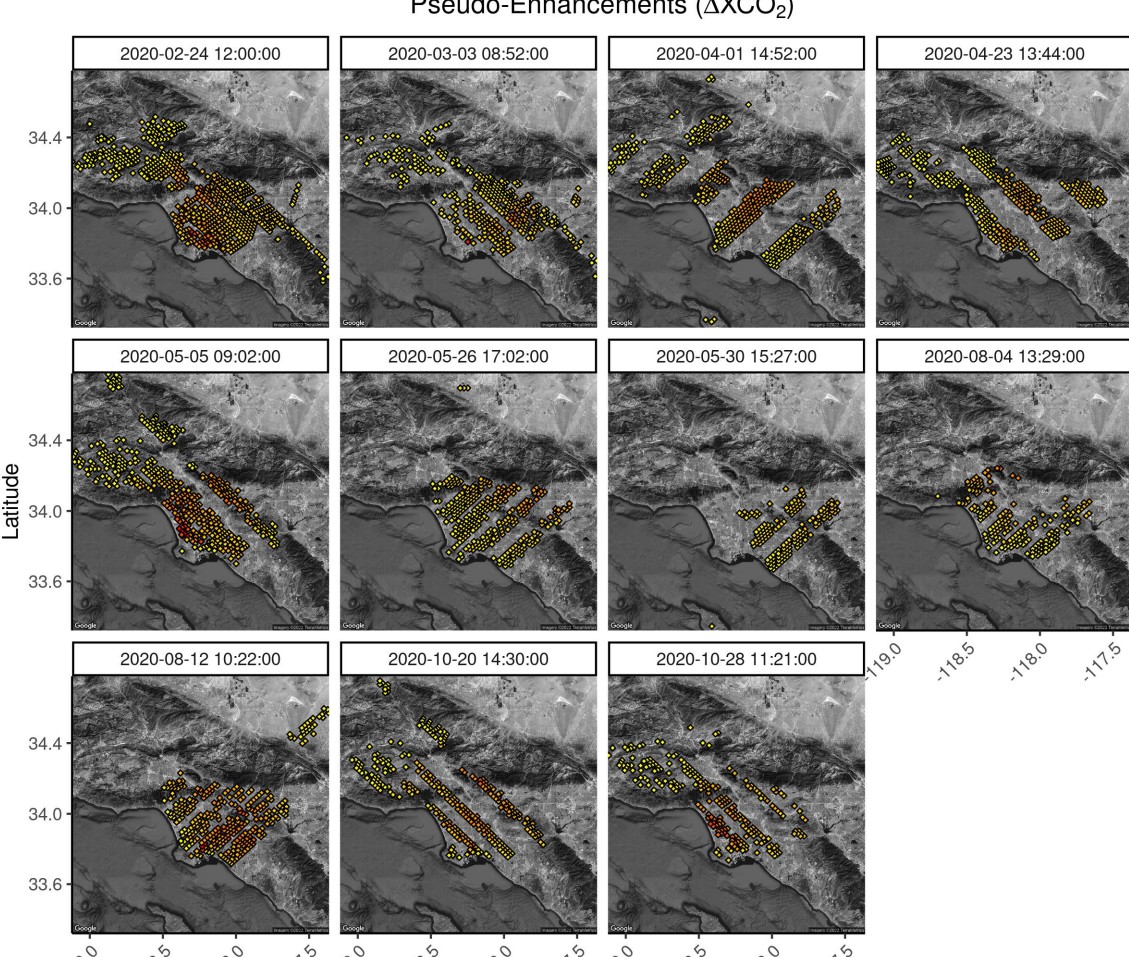

**Figure 3.** Enhancements of synthetic $XCO_2$ above regional background values are presented here. These 11 SAMs over the Los Angeles Basin were constructed using geolocated soundings from OCO-3 while $XCO_2$ enhancements were generated from the Vulcan 3.0 emissions dataset. Instrument error calculated from a previous study was included. The timestamps of the soundings making up each SAM were averaged to provide a single SAM observation time reported here in local time.

column of $s'_p$. Working under the assumption that a systematic multiplicative bias is present, **Eqn. 4** can be rewritten to include

a corrective scaling factor, $\lambda$, with length equivalent to the number of sectors under consideration (Lin and Gerbig, 2004):



$$\hat{\lambda} = \boldsymbol{\lambda}_p + \mathbf{S}_p \mathbf{K}^{\mathrm{T}} (\mathbf{K} \mathbf{S}_p \mathbf{K}^{\mathrm{T}} + \mathbf{R})^{-1} (\boldsymbol{z} - \mathbf{K} \boldsymbol{\lambda}_p). \tag{6}$$

A bias term, $\boldsymbol{\lambda}_p$, is a vector that represents a *prior* estimate of corrective factors, $\hat{\lambda}$. The uncertainty of these estimates are included along the diagonal of $\mathbf{S}_p$.

## 2.5 Test Cases

Applying the inversion to individual SAMs produced hourly *posterior* flux estimates for the 18 h prior to their observation times. These results were compared with corresponding *prior* and "*true*" flux fields to quantify the inversion's information content at both the per-SAM and hourly levels. Using four test cases, the sensitivity of these results to various input parameters was investigated. In three of the cases, outcomes varied with input grid resolution, atmospheric transport error, and flux error. The fourth test case investigated the SAMs' ability to correct systematic biases in *prior* flux estimates. These test cases are

described in the following sections and their titles and motivating questions are presented in **Tab. 2**.

**Table 2.** The titles and motivating questions associated with four OSSE test cases are presented here.

|  | **Motivating Questions** |
| --- | --- |
| **Test Case 1** | **Grid Size of Emission Inventories** |
|  | How does the coarsening of input grids affect uncertainty? |
|  | At what spatial resolutions can posterior emissions best be resolved? |
| **Test Case 2** | **Reduced Atmospheric Transport Error** |
|  | How does the use of more accurate atmospheric transport data affect the inversion scheme? |
|  | Will more accurate meteorology data decrease posterior uncertainty? |
| **Test Case 3** | **Well-Constrained Flux and Transport Error** |
|  | What if the uncertainty in prior flux estimates is well-constrained? |
|  | How will corrections in the locations of large point sources affect the inversion scheme? |
| **Test Case 4** | **Aggregated SAMs for Bias Correction** |
|  | Can multiple SAMs be aggregated in time and space to address systematic biases? |
|  | How many SAMs are necessary to fully correct biases in prior emissions estimates? |

### 2.5.1 Test Case 1: Grid Size of Emission Inventories

Similar to previous studies (Kunik et al., 2019; Mallia et al., 2020), this approach quantified *prior* uncertainty by incorporating the cell-level differences between the ODIAC-VIIRS and Vulcan 3.0 inventories during the construction of $\mathbf{Q}$ (details in **Sect. 2.7**). Although this *prior* error covariance matrix assigned an independent uncertainty to each flux value, inconsistently located

LPSs between inventories led to single cells with large values (see the difference plot in **Fig. 1** and Hogue et al., 2017). To mitigate these large single-cell errors, the input grids were coarsened to distribute these large magnitudes over a wider area;





thus, the first test case examined the ability of SAMs to optimize *prior* emissions estimated at a variety of input grid resolutions. Beginning with the highest possible resolution, 1 km $\times$ 1 km, the relationship between a SAM's ability to optimize a prior estimate and: (1) how many soundings were included within the SAM, (2) the time the soundings were collected, and (3) the strength of the observed enhancement were explored. In this case, *prior* fluxes were represented by ODIAC-VIIRS while Vulcan 3.0 represented the "*true*" flux. These inventories, along with influence footprints from X-STILT, were re-gridded to 3 km, 5 km, and 7 km resolutions but pseudo-observations of $\Delta XCO_2$ were consistently generated with the "*true*" 1 km gridding of Vulcan 3.0. **Fig. 4** presents the averaged spatial differences between all *prior* and "*true*" fluxes used in this work along with Moran's I values to quantify their spatial correlation. Accompanying p-values reveal decreasing statistical significance in spatial autocorrelation as the input grids are coarsened, revealing the loss of spatial structure.

### 2.5.2 Test Case 2: Reduced Atmospheric Transport Error

Earlier work presenting the X-STILT model (Wu et al., 2018) quantified uncertainty from atmospheric transport errors based on results driven by coarse meteorological inputs from the Global Data Assimilation System (GDAS). This $0.5° \times 0.5°$ input was responsible for an error $> 1$ ppm in $XCO_2$-space over the test case city of Riyadh, Saudi Arabia (Wu et al., 2018); however, this current study was conducted over LA, providing an opportunity to use the higher resolution HRRR meteorology (Benjamin et al., 2016; Rolph et al., 2017) or regionally tailored meteorological fields from the Weather Research and Forecasting (WRF) model (Powers et al., 2017). Although Test Case 1 (**Sect. 2.5.1**) used the prescribed error from Wu et al. (2018), the second test case used only high resolution input data and applied an error reduction of 50% to atmospheric transport (reduced magnitudes in the **R** covariance matrix; see **Sect. 2.7**). Comparisons between the original and reduced error scenarios were explored, revealing changes to *posterior* estimates from improvements in meteorological inputs.

### 2.5.3 Test Case 3: Well-Constrained Flux and Transport Error

Typically, the error in power plant geolocation databases within the United States is $< 1$ km (Woodard et al., 2014). Although there are several inconsistencies in the locations of LPSs displayed in **Fig. 1**, these can be explained by: (1) noting that the ODIAC-VIIRS and Vulcan 3.0 inventories are constructed for different years and (2) Vulcan 3.0 includes LPSs beyond those included in ODIAC-VIIRS. While Test Case 1 (**Sect. 2.5.1**) investigated the effects of these LPS-related errors, the third test case used a customized *prior* inventory to co-locate LPSs with the "*true*" inventory while maintaining the transport error reduction from Test Case 2. This scenario was representative of using a well-constructed emission inventory with accurate LPS information and high-quality meteorological information.

The customized *prior* was achieved by first identifying a threshold in which ODIAC-VIIRS cells of greater flux were mostly non-adjacent ($>75\ \mu mol\ m^{-2}\ s^{-1}$). These cells were replaced by values derived from averaging adjacent cells with the resulting array representing *prior* area emissions, $\mathbf{F}_{Prior,area}$. Sectors containing LPSs in the "*true*" emissions inventory were aggregated such that

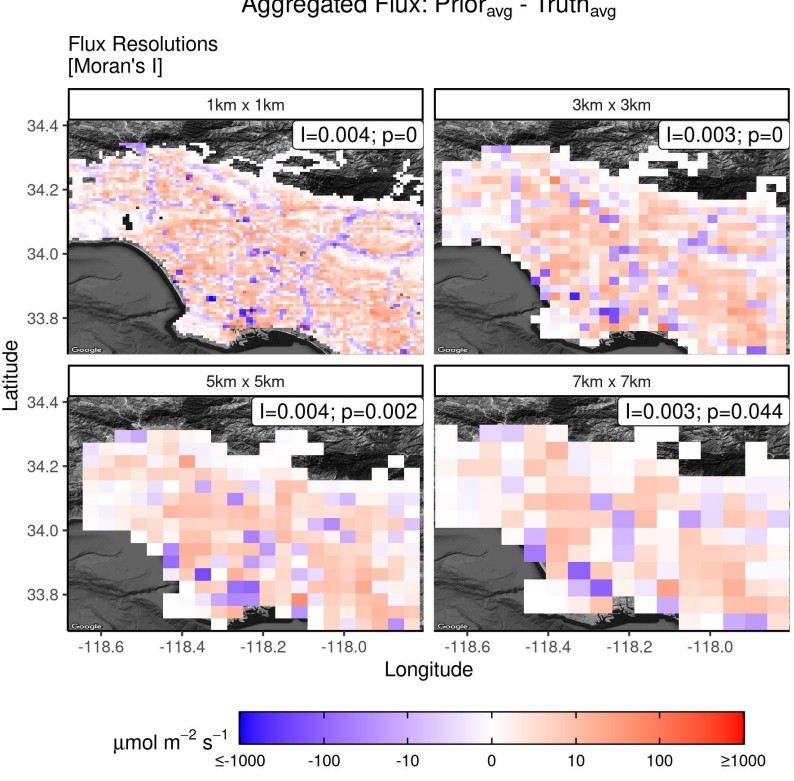

**Figure 4.** Presented here are the averaged differences between the ODIAC-VIIRS (*prior*) and Vulcan 3.0 ("*true*") emission inventories at four resolutions for the Los Angeles Basin. Also included are results from Moran's I tests to quantify the statistical significance ($p$) of the spatial correlation ($I$).

$$\mathbf{F}_{\mathrm{LPS}} = \mathbf{F}_{\mathrm{True,pwr}} + \mathbf{F}_{\mathrm{True,cem}} + \mathbf{F}_{\mathrm{True,avi}} \tag{7}$$

where the electricity production ($\mathbf{F}_{\mathrm{True,pwr}}$), cement manufacturing ($\mathbf{F}_{\mathrm{True,cem}}$), and commercial aviation ($\mathbf{F}_{\mathrm{True,avi}}$) sectors

were included. It was assumed that the area values in $\mathbf{F}_{\mathrm{Prior,area}}$ co-located with $\mathbf{F}_{\mathrm{LPS}}$ were from remaining area sources; therefore, these cells were scaled down before the inclusion.

The process of scaling of $\mathbf{F}_{\mathrm{Prior,area}}$ and adding LPSs was defined such that:

$$\hat{F}_{\mathrm{Prior},ij} = \left(1 - \frac{F_{\mathrm{LPS},ij}}{F_{\mathrm{True},ij}}\right) \cdot F_{\mathrm{Prior,area},ij} + F_{\mathrm{LPS},ij}. \tag{8}$$

Here, individual matrix elements were identified using $i,j$ notation. The ratio of each LPS flux to the "*true*" flux value ($F_{\mathrm{True},ij}$)

was used to determine the ratio of remaining area fluxes in Vulcan 3.0. This weighting ensured that LPSs added to the custom





inventory shared the same percentage of the cell's total. Results from using $\mathbf{F}_{\mathrm{True}}$ and $\hat{\mathbf{F}}_{\mathrm{Prior}}$ in the inversion scheme were used to explore its effectiveness under well-constrained error conditions.

The averaged differences between the customized *prior* and "*true*" inventories are presented in **Fig. 5** with original differences from the 1 km × 1 km case (**Sect. 2.5.1**). Grid cells containing large differences along the coast now demonstrate good

agreement. Results from a Moran's I test also suggests a stronger spatial correlation as the error is now driven by differences in area emissions. Notably, the large errors associated with the LPSs following the coastline are reduced.

## Aggregated Flux: Prior$_{\mathrm{avg}}$ - Truth$_{\mathrm{avg}}$

**Figure 5.** Presented here are the averaged differences between the modified *prior* and Vulcan 3.0 ("*true*") emission inventories for the Los Angeles Basin. For comparison, the unmodified case using ODIAC-VIIRS as the *prior* is included in the right panel. Results from Moran's I tests quantify the statistical significance ($p$) of the spatial correlation ($I$). Values in the range of $\pm 1\,\mathrm{\mu mol\,m^{-2}\,s^{-1}}$ are removed.

### 2.5.4    Test Case 4: Aggregated SAMs for Bias Correction

OCO-3 provides collections of SAMs for many locations and the constraint provided by multiple SAMs over a particular target is of interest. Therefore, the final case used **Eqn. 6** to explore the OCO-3 instrument's ability to correct systematic biases in

surface flux estimates. The effectiveness of the scaling factors provided by this process were simultaneously quantified for each emission category (**Tab. 1**). Beginning an iterative process, a vector of scaling factors ($\hat{\lambda}$) was calculated for a single SAM such that $\hat{\lambda} = \langle \hat{\lambda}_{\mathrm{pwr}}, \hat{\lambda}_{\mathrm{man}}, \hat{\lambda}_{\mathrm{trans}}, \hat{\lambda}_{\mathrm{build}} \rangle$. Subsequent calculations incorporated an additional SAM in succession. The results from




these iterative tests provided insights into sector-level sensitivity to $\Delta$XCO$_2$ as detected by the instrument. Two scenarios were considered: (1) the systematic bias was applied to all sectors, and (2) only area emissions were biased, assuming LPS values were well known.

### 2.6  Change in Relative Error: $\Delta\%_{\mathrm{rel}}$

A metric used throughout this work quantified the reduction in relative error after the inversion was applied. The absolute values of the differences between the *prior* and "*true*" emissions were compared to the *posterior* and "*true*" emissions such that:

$$\Delta\%_{\mathrm{rel}} = 100\% \times \left( \frac{|\mathrm{Posterior} - \mathrm{True}| - |\mathrm{Prior} - \mathrm{True}|}{\mathrm{True}} \right). \tag{9}$$

This calculation is used in both flux and XCO$_2$-space, where "Posterior" indicates optimized estimates in flux *or* XCO$_2$-space, "Prior" indicates the initial estimates of flux *or* $\Delta$XCO$_2$, and "True" indicates the exact values of surface flux *or* $\Delta$XCO$_2$. Results from this metric provided a spatially explicit quantification of improvements resulting from the inversion scheme. Negative values indicated the amount by which improvements were made while positive values indicated the amount by which the inversion moved the optimized estimate *further* from true values.

### 2.7  Sources of Error

There are two domains in the Bayesian inversion scheme where errors exist: XCO$_2$-space and flux-space. Errors in XCO$_2$-space were predominantly associated with atmospheric transport. The stochastic nature of the X-STILT model ($R_{\mathrm{part}}$), characterization of the planetary boundary layer ($R_{\mathrm{PBL}}$), and horizontal wind speeds ($R_{\mathrm{wind}}$) are sources of error that were considered. Wu et al. (2018) provided estimates of these values, and Kiel et al. (2021) reported the instrument error (inst.) in XCO$_2$ observations over the Los Angeles Basin. Since soundings away from the target domain are often used to calculate regional background XCO$_2$ values (bkg.), the same error was used for both $R_{\mathrm{inst}}$ and $R_{\mathrm{bkg}}$. The final source of influence considered in XCO$_2$-space was from the urban biosphere. To estimate potential errors arising from biospheric fluxes, X-STILT footprints generated for this study were convolved with the Solar-Induced Fluorescence for Modeling Urban Biogenic Fluxes (SMUrF) model which ingests biospheric observations to create spatially explicit net ecosystem exchange estimates (Wu et al., 2021). The hourly estimates provided by this model for the year 2019 were regridded to match the high resolution footprints (1 km × 1 km) and convolved to calculate the net contribution to each sounding (**Eqn. 2**). The standard deviation of the values was used as $R_{\mathrm{bio}}$. All XCO$_2$-space errors are included in **Tab. 3**.

Unlike the sparse spatial coverage of ground-based observation network sites, the OCO-3 instrument provides densely packed soundings at $\sim 2$ km spacing. This close proximity must incorporate the spatial correlation of observational errors in $\Delta$XCO$_2$. All XCO$_2$-space errors and their spatial correlation were represented in the inversion scheme (**Eqn. 4**) as error covariance matrix $\mathbf{R}$ where each element was defined as:





**Table 3.** Presented here are the error sources considered in $XCO_2$-space.

| Error Source | Variable | Error (ppm) | Citation |
|---|---|---|---|
| Stochastic Error | $R_{\mathrm{part}}$ | 0.06 | Wu et al. (2018) |
| Planetary Boundary Layer Height | $R_{\mathrm{PBL}}$ | 0.20 | " |
| Horizontal Wind | $R_{\mathrm{wind}}$ | 1.02 | " |
| Instrument | $R_{\mathrm{inst}}$ | 0.23 | Kiel et al. (2021) |
| Background | $R_{\mathrm{bkg}}$ | 0.23 | " |
| Biosphere | $R_{\mathrm{bio}}$ | 0.16 | Wu et al. (2021) |
| $\Sigma R_n^2 = 1.22\ \mathrm{ppm}^2$ | | | |

$$R_{ij} = \Sigma R_n^2 \cdot \mathrm{e}^{\frac{-X_{R,ij}}{l_R}}. \tag{10}$$

Individual errors were included as variances ($\Sigma R_n^2$) with the resulting value being scaled according to the distance matrix $\mathbf{X}_R$.

The correlation length scale, $l_R$, was calculated for each SAM by fitting an exponential semi-variogram to the differences between the soundings' *prior* $\Delta XCO_2$ estimates and pseudo-observations. Two correlation length scales were not calculated due a failure of convergence. As an alternative in these two instances, three times the minimum distance between soundings was used. The length scales are included in the far-right column of **Tab. 4** and demonstrate the variability across SAMs.

Errors associated with surface flux are represented by the $\mathbf{Q}$ error covariance matrix from **Eqn. 4**. In this OSSE, $\mathbf{Q}$ rep-
resented the differences between *prior* and "*true*" emissions while incorporating their spatial and temporal correlations such that:

$$\mathbf{Q} = \mathbf{I}_\sigma (\mathbf{D} \otimes \mathbf{E}) \mathbf{I}_\sigma. \tag{11}$$

Following the scheme presented by Kunik et al. (2019), differences between the emission inventories were represented by the diagonal matrix $\mathbf{I}_\sigma$ such that $I_{\sigma,ii} = |\bar{s}_{p,i} - \bar{s}_{t,i}|$ where $\bar{s}$ indicated an averaged emission vector across all fluxes used in this
work. The temporal and spatial covariances were represented by $\mathbf{D}$ and $\mathbf{E}$ respectively and were combined using a Kronecker product. Both covariance matrices are created using an exponential decay of the form:

$$M_{ij} = \mathrm{e}^{\frac{-X_{ij}}{l}}. \tag{12}$$

Here, $\mathbf{X}$ is a matrix relating values across time or space while the correlation length scale is represented by $l$. For temporal covariance, $\mathbf{X}_t$ was a square matrix containing the temporal differences between each backwards timestep associated with each
SAM's analysis. An autocorrelation function (ACF) was generated for each SAM by comparing the differences in total $CO_2$ between the *prior* and "*true*" inventories across the relevant timesteps. The number of significant timesteps ($p < 0.05$) from





the ACF dictated the value of $l_t$ for each SAM. To construct the spatial covariance matrix, the distances between the cells of the input grid were represented by the matrix $X_r$ and the spatial correlation length scales, $l_s$, were determined by applying an exponential semi-variogram fitting routine to the averaged spatial differences between the inventory timesteps. Each SAM's $l_t$
and $l_s$ values are reported in **Tab. 4** for the four test cases considered. The grid coarsening scenarios (Test Case 1, **Sec. 2.5.1**) are represented by their respective grid sizes (1-7 km), the case using co-located LPSs (Test Case 3, **Sec. 2.5.3**) is indicated by "Co-located Error", and the two different scenarios exploring the correction of systematic bias (Test Case 4, **Sec. 2.5.4**) are indicated by "LPS Error" and "No LPS Error". (Length scales for Test Case 2 are the same as Test Case 1.) The collection of $l_s$ and values associated with flux-space errors demonstrates less variability when compared to $l_R$. For this reason, mean $l_s$
values were used in each test case while SAM-specific $l_R$ values were used in the construction of each **R**. Similarly, the mode of the calculated $l_t$ values was applied to each test case.

A final calculation was needed to constrain the uncertainty associated with each *posterior* estimate. Since **Q** represented the cell-wise variance of the *prior* estimate, a calculation was performed to map cell-level **Q** values to posterior values such that

$$\mathbf{V}_{\hat{s}} = \mathbf{Q} - (\mathbf{HQ})^{\mathrm{T}}(\mathbf{HQH}^{\mathrm{T}} + \mathbf{R})^{-1}(\mathbf{HQ}) \tag{13}$$

where $\mathbf{V}_{\hat{s}}$ is the cell-level covariance associated with $\hat{s}$.

## 3 Results

### 3.1 Test Case 1: Grid Size of Emission Inventories

#### 3.1.1 Results in $XCO_2$-Space

As demonstrated in **Fig. 3**, variability in the coverage and observation times of SAMs is dictated by the orbit of the International
Space Station, changes in the viewing geometry, and cloud cover. Initial considerations given to SAM data quantified the effectiveness of the inversion based on observation times, the top 25% of "observed" $XCO_2$ enhancements per SAM, and the number of soundings making up each SAM. Relationships between these factors are presented in **Fig. 6**. Here, panel **(a)** contains a weak ($R^2 = 0.164$), yet statistically significant ($p < 0.05$) trend between the mean of upper quartile enhancements within each SAM and their observation times. The magnitudes of enhancements appear strong during the morning hours
but decrease throughout the day. This diurnal effect is attenuated due to the use of column-based observations rather than surface-based measurements. The deviations between co-located values in the plot are driven by the random error included in the pseudo-observations. Furthermore, **(c)** presents a significant trend ($p < 0.05$) between the amount of correction and the associated enhancement strength from each SAM. It demonstrates that the effectiveness of the optimization is directly proportional to the observed enhancement; however, as the input grid size increases ($1\mathrm{km} \times 1\mathrm{km}$ to $7\mathrm{km} \times 7\mathrm{km}$) there is more
variation in the relationship.





**Table 4.** The spatial and temporal length scales, $l_s$ and $l_t$ respectively, of the surface flux associated with each SAM is presented here. These values vary across test cases. The grid coarsening scenarios are represented by their respective grid sizes (1 km - 7 km), Large Point Source (LPS) Error indicates the uniform bias correction case, No LPS Error indicates the systematic bias scenario with well-constrained large point sources, and Co-located Error indicates the scenario in which a custom *prior* flux is used. The spatial correlation length scale of $XCO_2$ is included in the final column. (*) indicates SAMs where the fitting routine failed to converge, resulting in an assigned values of three times the minimum distance between soundings. The mean of the spatial correlation length scales and mode of the temporal autocorrelation length scales are included in the last row.

| | SAM Date | 1km | 3km | 5km | 7km | LPS Error | No LPS Error | Co-located Error | XCO$_2$ Error |
|---|---|---|---|---|---|---|---|---|---|
| | | | | | $l_s$ [km]; $l_t$ [h] | | | | $l_R$ [km] |
| 1 | 02/24 | 2.56; 2 | 1.79; 2 | 3.00; 2 | 9.21; 2 | 2.14; 3 | 4.31; 3 | 1.68; 2 | 7.16 |
| 2 | 03/03 | 2.48; 2 | 2.00; 2 | 2.95; 7 | 9.17; 2 | 2.00; 3 | 3.96; 3 | 1.52; 2 | 4.13 |
| 3 | 04/01 | 2.39; 2 | 1.89; 2 | 3.48; 2 | 8.77; 2 | 2.32; 3 | 3.75; 3 | 1.31; 3 | 5.00* |
| 4 | 04/23 | 2.63; 2 | 2.10; 3 | 3.00; 2 | 8.50; 2 | 2.27; 3 | 3.55; 3 | 1.46; 3 | 1.87 |
| 5 | 05/05 | 2.45; 2 | 1.92; 2 | 3.39; 2 | 9.10; 3 | 2.11; 3 | 3.64; 3 | 1.47; 2 | 3.58 |
| 6 | 05/27 | 2.29; 2 | 1.48; 2 | 2.92; 2 | 8.63; 7 | 2.42; 3 | 3.39; 3 | 1.23; 2 | 7.61 |
| 7 | 05/30 | 3.44; 2 | 1.59; 3 | 3.59; 3 | 8.80; 2 | 2.65; 4 | 3.20; 4 | 1.31; 3 | 6.00* |
| 8 | 08/04 | 3.07; 2 | 1.68; 2 | 1.80; 7 | 8.86; 2 | 1.79; 3 | 3.33; 3 | 1.70; 2 | 0.66 |
| 9 | 08/12 | 2.17; 2 | 1.59; 2 | 0.29; 3 | 8.86; 2 | 1.31; 3 | 3.44; 3 | 1.66; 2 | 8.28 |
| 10 | 10/20 | 3.09; 2 | 1.85; 3 | 3.02; 3 | 9.37; 2 | 1.99; 3 | 3.61; 3 | 1.53; 3 | 5.12 |
| 11 | 10/28 | 3.21; 3 | 1.94; 3 | 6.44; 3 | 8.38; 2 | 1.82; 3 | 3.77; 3 | 1.49; 3 | 3.28 |
| | Mean of $l_s$; Mode of $l_t$ | 2.71; 2 | 1.80; 2 | 3.08; 2 | 8.88; 2 | 2.07; 3 | 3.63; 3 | 1.50; 2 | |

Conversely, panel **(b)** of **Fig. 6** demonstrates no clear trends between the amount of correction and the number of soundings making up each SAM ($p > 0.05$) nor do patterns exist across grid resolutions. Additionally, no clear trends exist between corrections from SAMs and the time that they were observed (**[d]**), indicating that the weak trend of decreasing enhancement throughout the day does not significantly affect the inversion's ability to optimize estimates.

A key driver of the inversion scheme (**Eqn. 4**) is the initial difference between *prior* $\Delta XCO_2$ estimates ($\mathbf{H}s_p$) and the co-located pseudo-enhancements ($z$); therefore, the individual soundings that comprise each SAM were aggregated and investigated collectively. These soundings are presented in the top panels of **Fig. 7**. Here, the per-sounding optimization, driven by $|z - \mathbf{H}s_p|$, is plotted with a running average and linear regression. Included in the bottom set of panels is a collection of spatial plots where all $\mathbf{H}s_p$ and $z$ differences were superimposed and averaged at a $0.01° \times 0.01°$ resolution.

The regressions included in the top panels of **Fig. 7** (blue lines) demonstrate the inversion's progressively decreasing ability to optimizatize *prior* estimates as input grid size increases. The slope of the linear fit in the high resolution case ($1km \times 1km$) suggests that only $54\%$ of the per-sounding initial difference is mapped to the error reduction in *posterior* estimates of $\Delta XCO_2$.

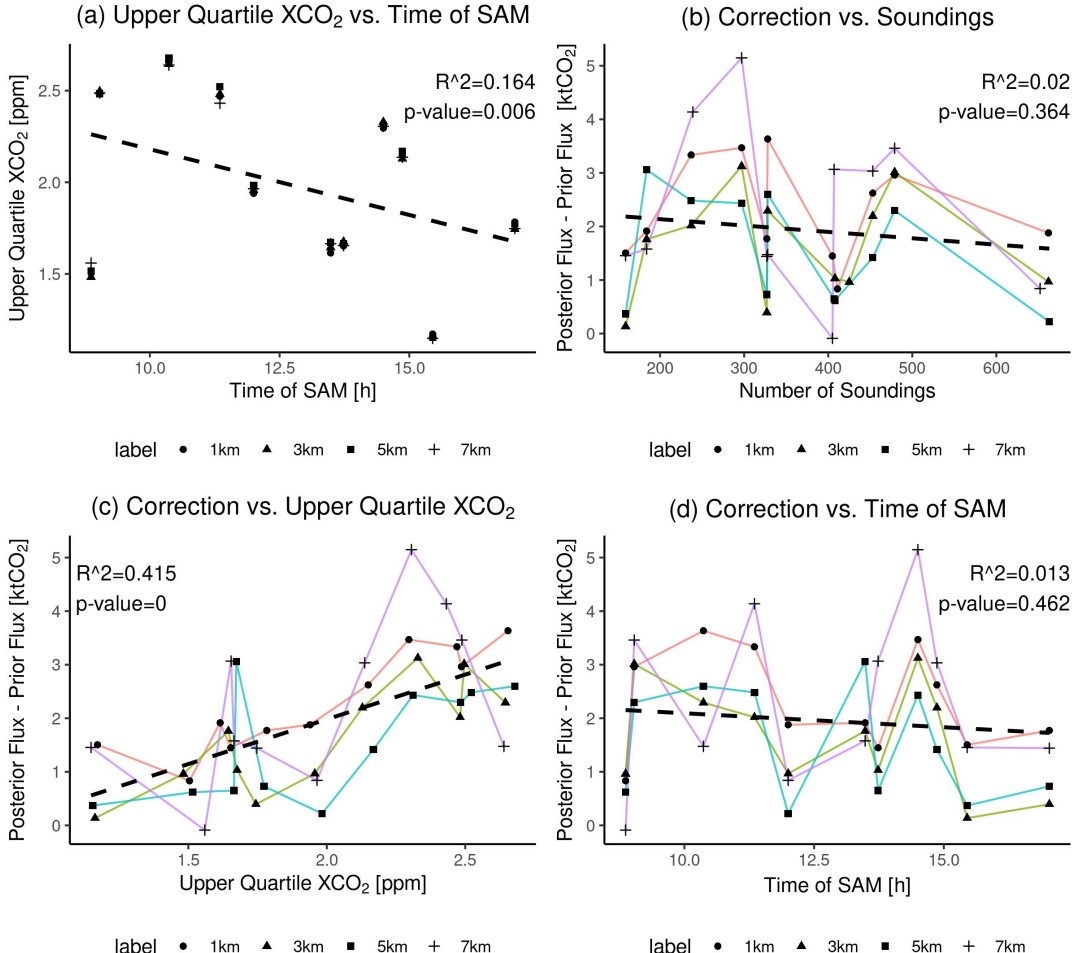

**Figure 6.** (a) The mean of the highest 25% enhancement values from each SAM is plotted against their local observation time. Slight deviations in co-located values are present due to random observational error being introduced for each re-gridding scenario. (b) The amount of correction provided in kilotons of $CO_2$ is plotted against the number of soundings making up the associated SAM. Results from each re-gridding scenario are included. (c) The amount of correction provided by the inversion is plotted against the mean of the highest 25% enhancement values associated with each SAM. These values are included for all input grid resolutions. (d) The amount of correction provided by the inversion is plotted against the time of day (local time) under consideration. Data from all re-gridding scenarios are included. In all four panels, a single linear regression is included (black, dashed line) along with its $R^2$ and p-value.

This value is inversely proportional to the input grid size, with only 15% mapped to error reductions at the most coarse scale. Furthermore, the non-linearity of the running averages does not show an increase in correction until roughly 0.5 ppm, favoring larger initial differences. As the resolution of input data decreases, it is clear that the *posterior* estimates' sensitivity to initial differences follows.




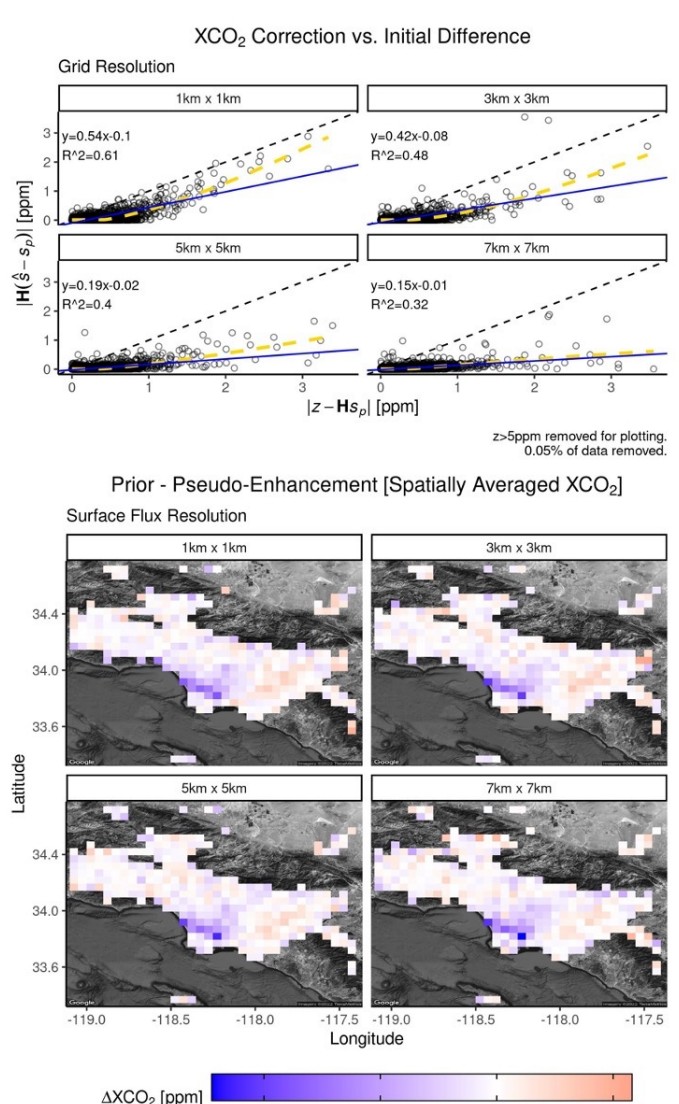

**Figure 7.** [Top] The per-sounding differences between *prior* and *posterior* estimates of $\Delta XCO_2$ are plotted against the differences between *prior* and pseudo-enhancements of $\Delta XCO_2$ for multiple input grid resolutions. The 1:1 line is represented in black. An ordinary linear regression (blue) and running average (yellow) are also included. The equation of linear fit and $R^2$ values for each regression are included. [Bottom] The per-sounding differences between *prior* and pseudo-$\Delta XCO_2$ across all SAMs are superimposed and averaged at a $0.01° \times 0.01°$ resolution over the Los Angeles Basin. Four different resolutions of underlying fluxes are represented by the panels.

The results in **Fig. 7 (top)** demonstrate that the inversion's effectiveness is dependent on larger initial differences; therefore, it is crucial to understand *where* these large differences occur and *what* drives them. Spatially averaged initial differences are presented in **Fig. 7 (bottom)** and demonstrate a locational dependence along the southwest coast and in the eastern part of the





basin. Along the coast, *prior* $\Delta XCO_2$ estimates are smaller than pseudo-enhancements. Conversely, *prior* $\Delta XCO_2$ estimates are larger than pseudo-enhancements to the east. When comparing the differences across progressively coarsened input grids, the coastal underestimation becomes more profound while the eastern overestimation is reduced. These large initial differences along the coast are driven by the presence of LPSs in the Vulcan 3.0 inventory that are missing in ODIAC-VIIRS and are reduced during the grid coarsening process (see **Fig. 4**). The difference plots in **Fig. 7 (bottom)** and non-linear dependence

**(top)** suggest that the inversion scheme is effective along the coast where the initial differences in $XCO_2$-space are the highest across all input grid resolutions.

  To quantify the spatial dependence of the inversion scheme's effectiveness, differences between aggregated *posterior* and *prior* $\Delta XCO_2$ are plotted in **Fig. 8 (left)**, showing strong corrections >0.5 ppm along the coast and further validating the sensitivity of the inversion to this region. The spatial distribution of *posterior* $\Delta XCO_2$ correction is strongly correlated with the

*prior* underestimation shown in **Fig. 7**; however, overestimates to the east are not corrected due to their small initial differences (<0.5 ppm). Although the initial differences along the coast increase as the input grid resolution coarsens, the amount of change from the *posterior* is inversely proportional to the input grid size due to cell-wise decreases in flux uncertainty at coarse resolutions. To ensure that the increases in $\Delta XCO_2$ provided by the *posterior* are reasonable, $\Delta\%_{rel}$ is calculated for these results (see **Sect. 2.6**). These aggregated per-sounding changes in relative error (*prior* to *posterior*) are presented in **Fig. 8**

**(right)** and show strong reductions in $\Delta XCO_2$ error (∼25%) along the coast. Here, optimizations that are smaller in magnitude are more visible than in *posterior* vs. *prior* comparisons. As expected, there are smaller reductions in relative error as the input grids become more coarse.

### 3.1.2 Cumulative Results in Flux-Space

Although relative errors in $XCO_2$-space were reduced, the *posterior* estimate produced by the inversion may not provide

equivalent error reductions in flux-space. To investigate flux-space errors, the 18 h spans of *prior*, *true*, and *posterior* fluxes ($\mu mol\,m^{-2}\,s^{-1}$) associated with each SAM were used to calculate the total $CO_2$ emitted from the urban domain of the LA Basin during each timeframe (see **Fig. 1**). Ideally, the differences between total *posterior* and "*true*" values will be smaller than the differences between *prior* and "*true*" values. These differences are plotted, relative to the "*true*" total, in **Fig. 9** and reveal that *posterior* emissions are systematically overestimated. All occurrences of the *posterior* difference being larger than

the *prior* difference are indicated by red outlines in the plot. The only SAMs demonstrating improvements are those where the *prior* total is lower than the "*true*" total (SAMs 3, 6, 7, and 11), allowing for the systematic overestimation to approach the "*true*" value. This resulting overestimation holds across the input grid sizes considered.

  The error bars included in **Fig. 9** were calculated from $\mathbf{Q}$ and $\mathbf{V}_{\hat{s}}$. For the 18 hourly timesteps associated with each SAM, the total emitted $CO_2$ from the *prior* and *posterior* uncertainties were added in quadrature, serving as the reported error for the

emission totals. These errors appear to be independent from the resolution of the input grid as the $3\,km \times 3\,km$ results have the smallest error; however, the magnitudes follow the values of $\bar{l}_s$ as reported in **Tab. 4** where the 3 km case has the smallest length scale and the 7 km case has the largest.

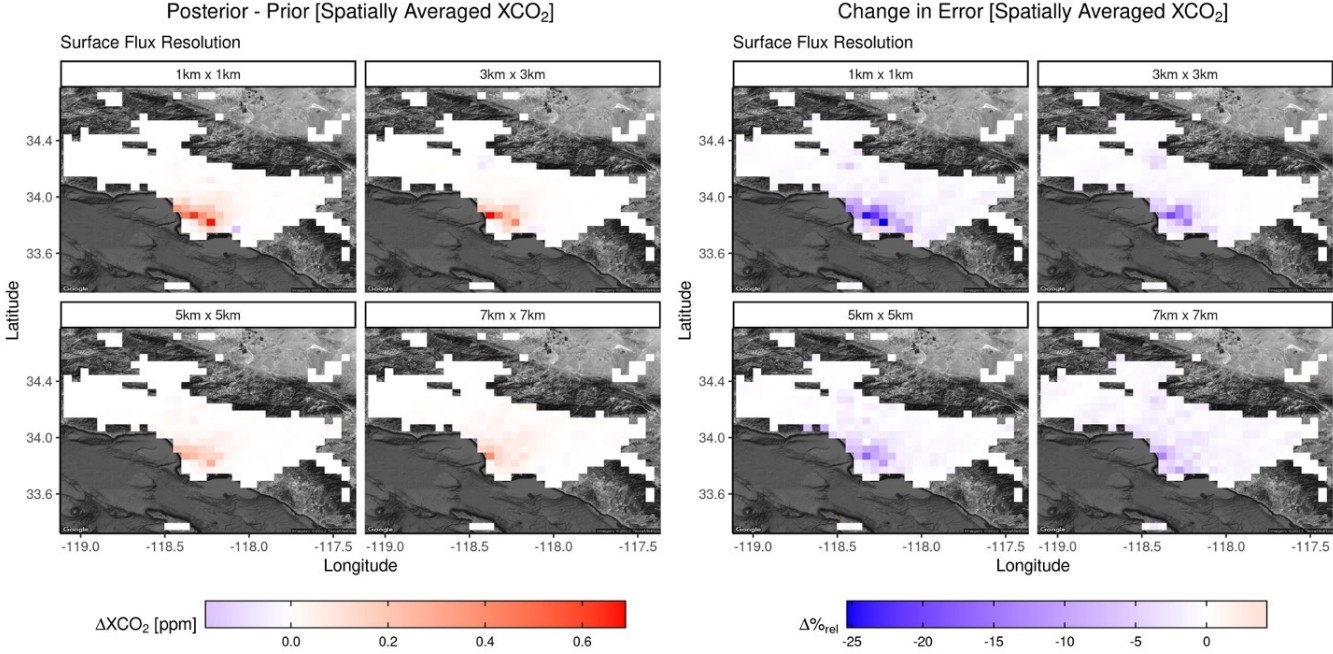

**Figure 8.** [Left] Per-sounding post-inversion changes in $\Delta XCO_2$ for all SAMs were superimposed and averaged at a $0.01° \times 0.01°$ resolution over the Los Angeles Basin. The results are included for each grid resolution of interest. [Right] Per-sounding changes in the relative error of $\Delta XCO_2$, also aggregated as described (left), are plotted for all input grid resolutions of interest.

### 3.1.3  Hourly Results in Flux-Space

Results presented in **Figs. 8** and **9** show that the inversion reduces errors in $XCO_2$-space while sacrificing accuracy in flux-
space. The systematic overestimation in total $CO_2$ was driven by increasing *prior* flux magnitudes more than required. To gain more insight, the summed contributions in **Fig. 9** were disaggregated in time to determine *when* overestimations occur.

At the hourly level, the Bayesian inversion's ability to optimize each timestep depends on the magnitude of footprint values coincident with the domain during the timestep in question. Although nearly all footprints start within the domain for the first backwards timestep, many are fully removed due to atmospheric transport after several hours. The information density of each
SAM is presented in **Fig. 10** where the footprint-weighted average amount of hourly correction (in metric tons (mt) of $CO_2$) is provided for each SAM. The initial backwards timestep contains the most information density while the magnitudes of consecutive steps decay sharply. Linear regressions applied to log-log transformations of the data are included (black, dashed lines) along with their slopes and y-intercepts. The y-intercepts indicate that the average maximum information density of the SAMs is inversely proportional to the input grid size. The 1 km resolution case has an average maximum information density
of $17.83 \, \mathrm{mtCO_2}$ while the intermediate cases have values $7.48 \, \mathrm{mtCO_2}$ and $6.98 \, \mathrm{mtCO_2}$. The lowest value, $6.43 \, \mathrm{mtCO_2}$, occurred at the most coarse resolution. At a high resolution, the inversion's ability to optimize a *prior* estimate follows a



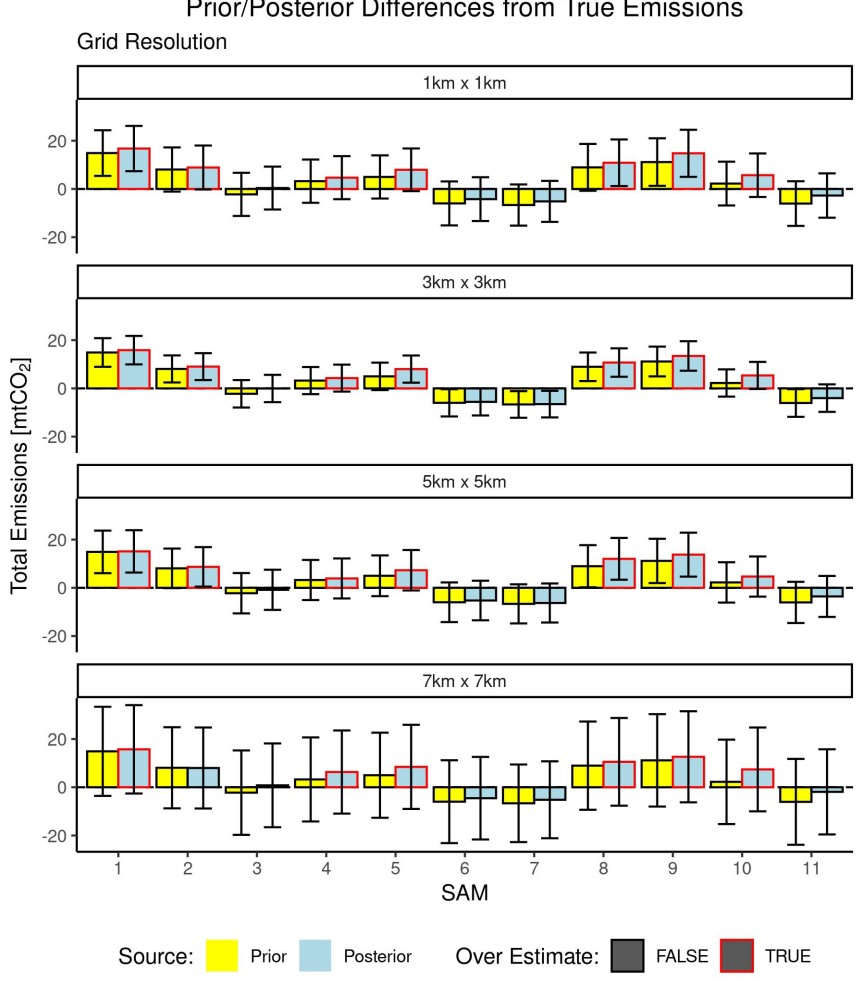

**Figure 9.** Differences in the total $CO_2$ emitted (in metric tons) over the 18 h timeframes associated with each SAM are presented here. *Prior* and *posterior* estimates were compared against the "*true*" total. Cases where |*posterior - true*| is larger than |*prior - true*| are indicated with red borders. Error bars represent error derived from the *prior* and *posterior* covariance matrices.

$(\Delta t)^{-2.3}$ distribution while the value is decreased to $(\Delta t)^{-1.62}$ in the coarse resolution suggesting coarser resolution input data can be optimized further backward in time. The hourly corrections presented in **Fig. 10** suggest that the excessive addition of $CO_2$ to the *prior* estimates occurs during the first few hours of the inversion.

Spatial distributions of the hourly changes in error are presented in **Fig. 11**, where $\Delta\%_{rel}$ was calculated for the fluxes associated with each SAM to determine *where* overestimations occur. The results were superimposed and aggregated across all SAMs, only stratified by hourly backwards timestep. This averaging revealed that the inversion scheme increased relative flux error along the coast when compared to the initial error in the *prior* estimate. Clusters of error reductions, although smaller

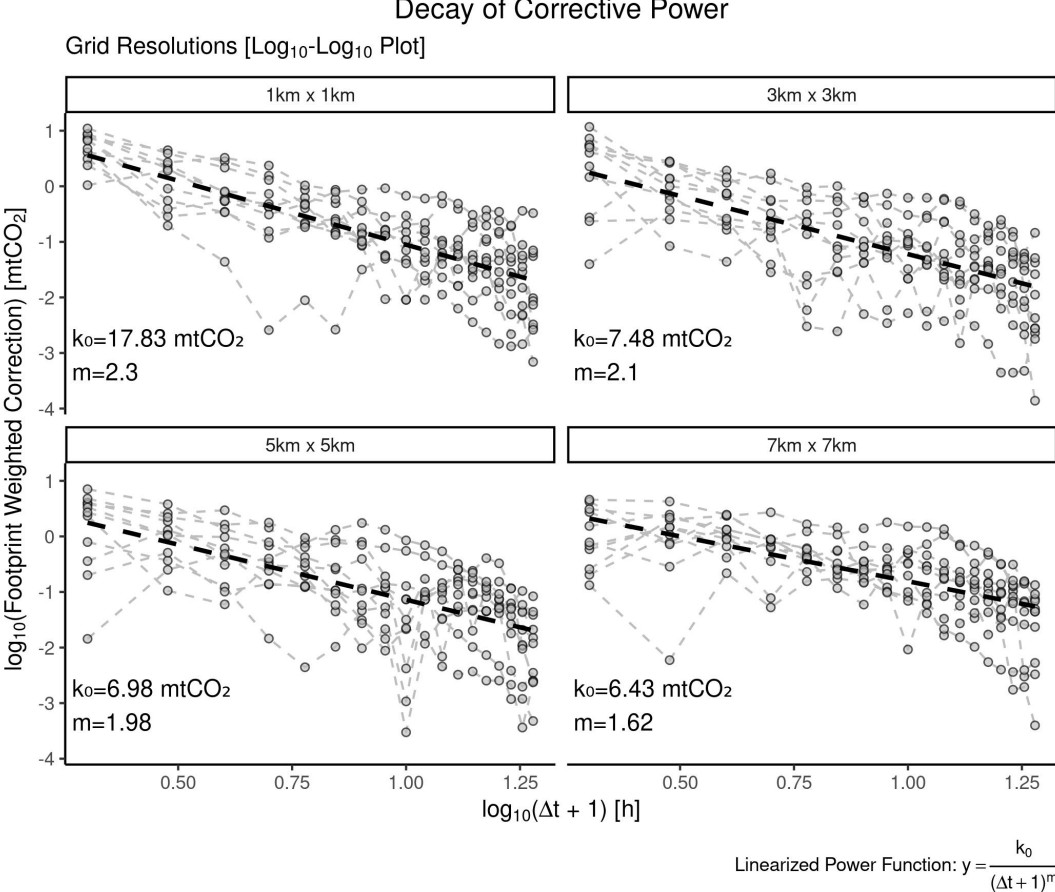

**Figure 10.** The footprint-weighted mean correction (in $\mathrm{mtCO_2}$) applied to each backward timestep has been plotted for each SAM. Timestep values associated with each SAM are connected by gray dashed lines. These values are displayed in a log/log plot with a linear fit. The resulting y-intercepts ($k_0$) and slopes ($m$) of these regressions are reported in each panel. Plots for multiple input grid resolutions are displayed.

in magnitude, are present to the north and east. At the highest resolution, clusters of increased error (positive values) appear

to have nuclei of cells where error reduction occurred. These isolated cells of reduced error correspond to LPSs in the Vulcan 3.0 emission inventory (see **Fig. 1**). During the construction of **Q**, these cells were assigned large uncertainties due to LPS mismatch between inventories. They dominate the corrective power of the inversion, requiring a large change in the *prior* flux value to match the value of "*true*" flux. The spatial correlation among neighboring cells forces a similar, yet unwarranted, change in flux values. Conversely, areas without nearby LPSs experience post-inversion reductions in relative error.

Proceeding through the averaged changes in relative error, the high-resolution case reveals that initial increases in flux error near the coast decay yet are present through almost all of the 18 timesteps. Conversely, the areas of error reduction to the east



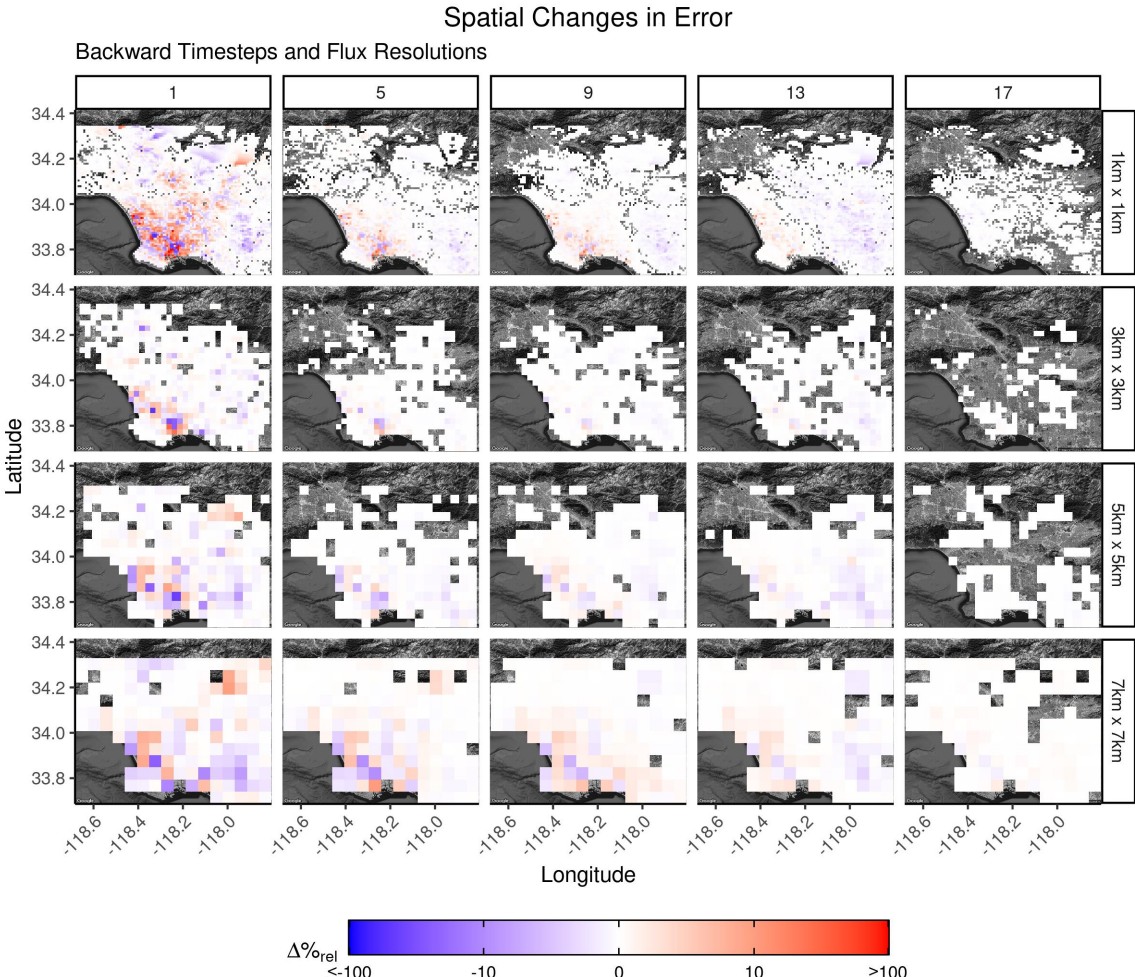

**Figure 11.** Presented here are the averaged changes in relative flux error provided by the inversion scheme. Backward timesteps associated with 11 SAMs were superimposed and averaged for the Los Angeles Basin. Columns indicate particular hourly timesteps from the time of observation and rows indicate the spatial resolution of the input data. Small values in the range ±0.01% are excluded.

decay quickly. As the input grids become coarser, the magnitudes of the changes in error are reduced, the area of increasing error is sustained for more timesteps, and the area of error reduction to the east is still present. The results of **Fig. 11** suggest that the systematic overestimation of $CO_2$ (see **Fig. 9**) occurs during the first few backwards timesteps of the inversion near

LPS locations. As the input resolution coarsens, the effect increases in duration.





### 3.2 Test Case 2: Reduced Atmospheric Transport Error

The transport error used in Test Case 1 was $R_{\mathrm{wind}} = 1.02 \, \mathrm{ppm}$. Under the assumption that alternative meteorological datasets would provide better representations of transport, the benefits of a 50% reduction in transport error were explored. **Fig. 12** compares aggregated $\Delta \mathrm{XCO_2}$ information by first examining the per-sounding amount of correction made available by initial differences. In the previous test case, 100% transport error allowed for $\sim$54% of the initial difference in $\Delta \mathrm{XCO_2}$ to be mapped to corrections; however, the running average revealed that soundings with the largest differences received the most benefit. This was also the case when $R_{\mathrm{wind}}$ was reduced by 50%. This error reduction allowed for subtle increases in corrections from smaller initial differences, also shown by the sharp increase in the running average in the top right-hand panel. Overall, the linear regression suggests that $\sim$67% of these differences are applied to corrections in per-sounding $\Delta \mathrm{XCO_2}$, but only the largest initial differences see the most benefit.

Spatially, the per-sounding corrections in $\mathrm{XCO_2}$-space were again aggregated at a $0.1° \times 0.1°$ resolution, revealing similar results to the previous case. The numerator of **Eqn. 9** was calculated using $\Delta \mathrm{XCO_2}$ pseudo-observations and compared to results from the previous section (bottom panels of **Fig. 12**). This comparison shows that reductions in $\Delta \mathrm{XCO_2}$ error were more intense when the atmospheric transport error was reduced and the spatial distribution of corrections was increased to include areas north of the coast.

Again, results in $\mathrm{XCO_2}$-space show better agreement with "*true*" and pseudo-observation values; however, post-inversion optimizations in flux-space reveal larger discrepancies between *posterior* and "*true*" estimates. Presented in **Fig. 13** are totals of emitted $\mathrm{CO_2}$ from the domain during the timeframe associated with each SAM. Compared to the previous case, the differences between estimated totals and the "*true*" value reveals greater deviations when the transport error was reduced. Increased differences are indicated with red borders. As in the previous test case, the "*true*" total lies within the error bars of several *prior* and *posterior* estimates. Error bars were calculated using the method described in **Sec. 3.1.2**.

As in the previous test case, the larger overestimates of total $\mathrm{XCO_2}$ generated by the reduction in atmospheric transport can be explained by temporally disaggregated results. **Fig. 14** presents averages of the backward timesteps associated with the SAMs used in this study. Results from the reduced error case are included along with results from the previous case. The bottom row of panels show that a 50% reduction in atmospheric transport error drives further increases in relative flux error along the coast. In the first backward timestep of the inversion, the area where relative errors increase is larger; however, reduced transport error shows better inversion performance in the east part of LA. The areas of increasing error dominate any reductions, systematically increasing the $\mathrm{CO_2}$ emitted. Both areas of increasing and decreasing error are better sustained across the 18 h timesteps in the reduced transport error case.

### 3.3 Test Case 3: Well-Constrained Flux and Transport Error

Results from the second test case (**Sect. 3.2**) demonstrated that reductions in atmospheric transport error alone increased the magnitude and area of $\mathrm{XCO_2}$-space optimization; however, these corrections were accompanied by increased flux error. Presented here are results from the inversion scheme under the assumption that both atmospheric transport and LPS-related

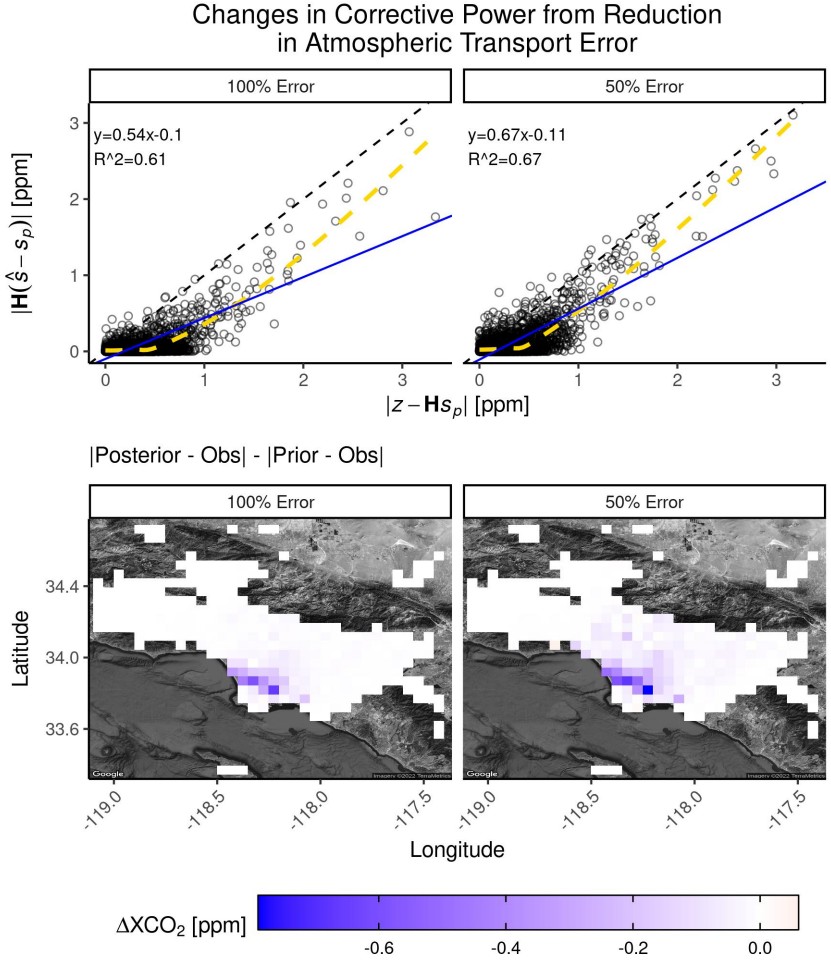

**Figure 12.** [Top] The per-sounding differences between *prior* and *posterior* estimates of $\Delta XCO_2$ are plotted against the differences between *prior* and pseudo-enhancements of $\Delta XCO_2$ for two different values of atmospheric transport error. The 1:1 line is represented in black. An ordinary linear regression (blue) and running average (yellow) are also included. The equation of linear fit and $R^2$ values for each regression are included. [Bottom] The per-sounding differences between *prior* and pseudo-$\Delta XCO_2$ across all SAMs are superimposed and averaged at a $0.01° \times 0.01°$ resolution over the Los Angeles Basin. Both cases of originally reported and reduced transport error are included.

flux uncertainties are well known. The transport error used here remained at 50% and the customized *prior* flux, generated by
co-locating LPSs (**Sect. 2.5.3**), served as the *prior* flux estimate, allowing for smaller uncertainties.

The co-location of LPSs reduced differences in *prior* and "observed" $\Delta XCO_2$. This is shown in the top of **Fig. 15** where the per-sounding corrections are plotted against their associated initial differences, with results from the first test case also included for comparison. Previously, results demonstrated that only larger initial differences are capable of driving optimization due to the large errors ($\mathbf{R}$) and small signals associated with $\Delta XCO_2$ enhancements. In the case of colocated LPSs, a majority



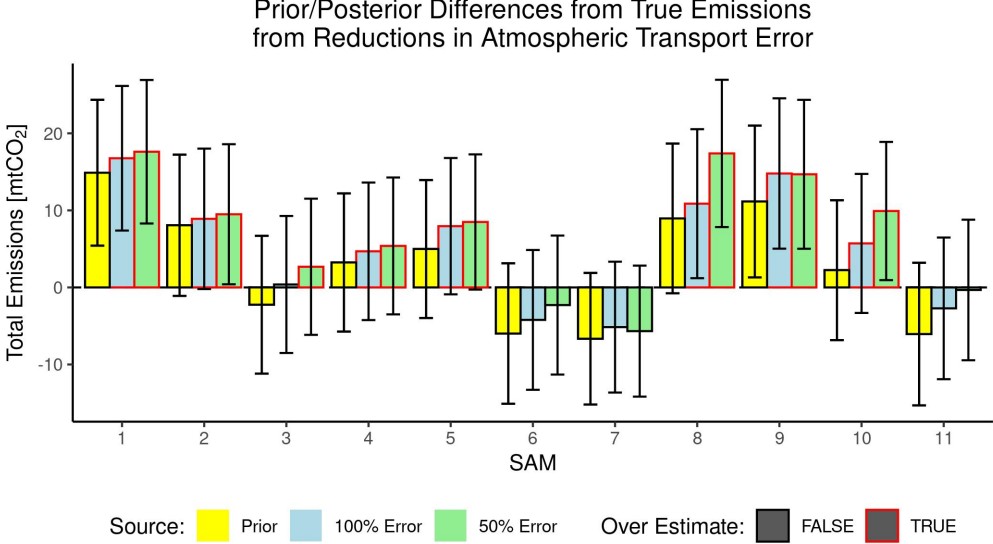

**Figure 13.** The hourly fluxes influencing each SAM were used to determine the total $CO_2$ emitted from the domain. *Prior* and *posterior* estimates were compared against the "*true*" amount emitted. Cases where |*posterior - true*| is larger than |*prior - true*| are indicated with red borders. Error bars represent error derived from the *prior* and *posterior* covariance matrices. Here, *posterior* estimates gathered from two different transport error values are plotted.

of the initial differences lie below 1 ppm. In the first test case where LPS locations were inconsistent, roughly 54% of the initial differences are mapped to corrections in *posterior* $\Delta XCO_2$ values; however, the removal of the main driver of initial differences provides only 4% as corrections. The bottom panels of **Fig. 15**, which show reductions in absolute error over LA, demonstrate localized improvements near the locations of LPS mismatch. When these mismatched cases are resolved corrections are removed.

Although the inversion's ability to optimize *prior* estimates is no longer visible in $XCO_2$-space, *posterior* flux estimates show improvement. Both the modified *prior* and *posterior* flux estimates were used to calculate the total $CO_2$ emitted during each SAM's associated inversion timeframe (18 h leading up to the SAM observation). These totals were compared with the "*true*" total emissions and the differences are presented in **Fig. 17**. In the top panel of the figure, results from the first test case are included which show a systematic addition of excess $CO_2$. After reducing $R_{wind}$ by 50% and constraining LPS locations,

the overestimation is no longer present and *posterior* totals are slightly closer to "*true*" totals.

   There are a few notable features in the bottom of **Fig. 16**. When comparing results to the previous test case, larger differences between the *prior* and "*true*" total emissions are evident. During the creation of the custom flux, LPS values are added to background emissions from the ODIAC-VIIRS inventory. As shown in **Fig. 1**, this inventory has higher area emissions than Vulcan 3.0. The addition of more LPS values beyond the power plants included in ODIAC-VIIRS further raises the total



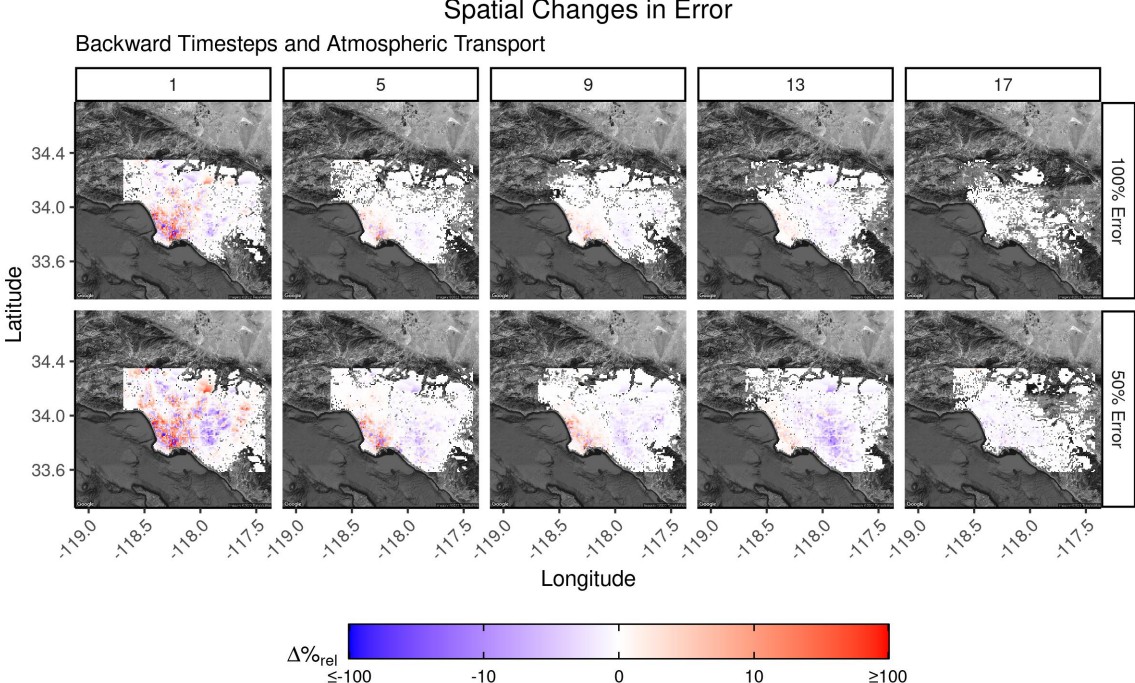

**Figure 14.** Presented here are the averaged changes in relative flux error provided by the inversion scheme. Backward timesteps associated with 11 SAMs were superimposed and averaged for the Los Angeles Basin. Columns indicate particular hourly timesteps from the time of observation and rows indicate the value of atmospheric transport error used. Small values in the range $\pm0.01\%$ are excluded.

emissions of the custom *prior*. Additionally, the error bars included in the custom *prior* comparisons are smaller than those in the first test case (top). This is driven by a smaller spatial length scale ($\bar{l}_s = 1.5$ km; see **Tab. 4**).

The improvements to total emission estimates found in **Fig. 16** were accompanied by similar improvements in flux estimates. Results from calculating $\Delta\%_{\mathrm{rel}}$ at hourly timesteps and averaging across all SAMs are presented in **Fig. 17**. Here, the first four backwards timesteps are presented from the original *prior* flux (top row) (**Sect. 3.1.2**) and the customized flux (bottom row). Under well-constrained conditions where atmospheric transport error was reduced and LPS locations were consistent, relative error reductions were clustered throughout LA. Although some areas experienced increased error, the large area of biased optimization found in the unmodified flux case was no longer present along the coast. Also apparent was the sharp decay in both increases and decreases in error. After the initial backwards timestep, no clusters of significant change existed within the customized flux. This is counter to the unmodified case where large flux error biases were maintained along the coast for several backwards timesteps.



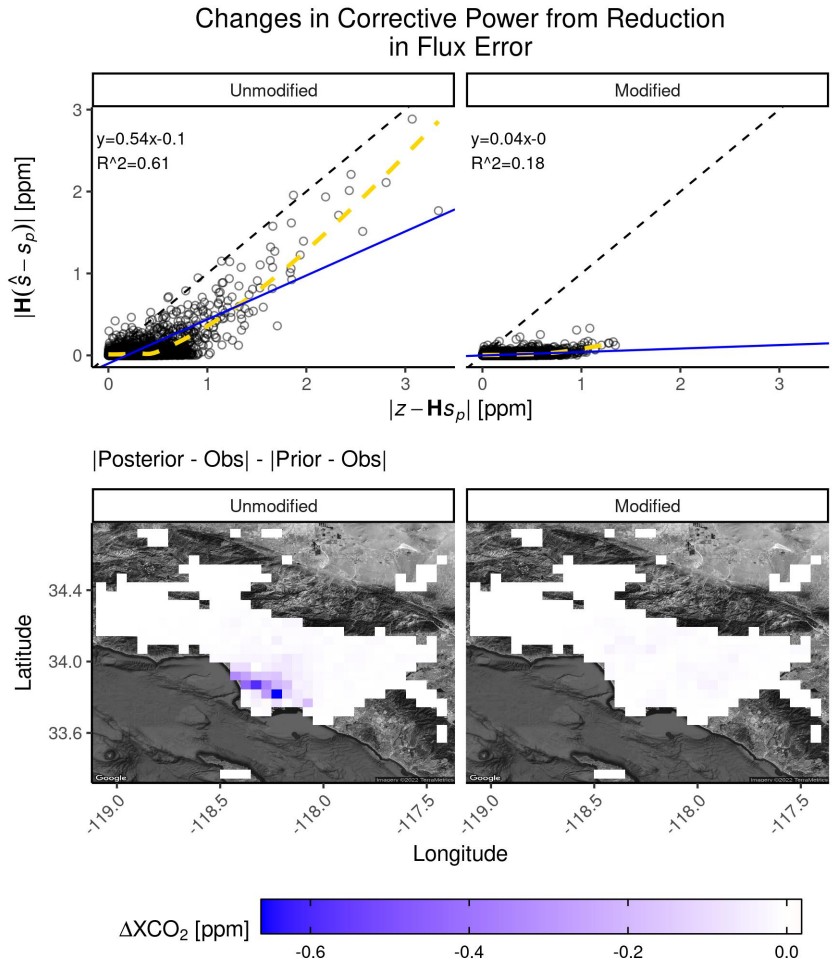

**Figure 15.** [Top] The per-sounding differences between *prior* and *posterior* estimates of $\Delta XCO_2$ are plotted against the differences between *prior* and pseudo-enhancements of $\Delta XCO_2$ for two different configurations of *prior* flux. "Unmodified" flux values include ODIAC-VIIRS estimates while "Modified" flux values include area flux values from ODIAC-VIIRS and "*true*" LPS values from Vulcan 3.0. The 1:1 line is represented in black. An ordinary linear regression (blue) and running average (yellow) are also included. The equation of linear fit and $R^2$ values for each regression are included. [Bottom] The per-sounding differences between *prior* and pseudo-$\Delta XCO_2$ across all SAMs are superimposed and averaged at a $0.01° \times 0.01°$ resolution over the Los Angeles Basin. Both cases of modified and unmodified *prior* are included.

## 3.4 Test Case 4: Aggregated SAMs for Bias Correction

The previous sections considered the constraints on fluxes provided by individual SAMs while the final test case utilized the information content from multiple SAMs to correct systematic biases in sector-specific information. The emissions categories



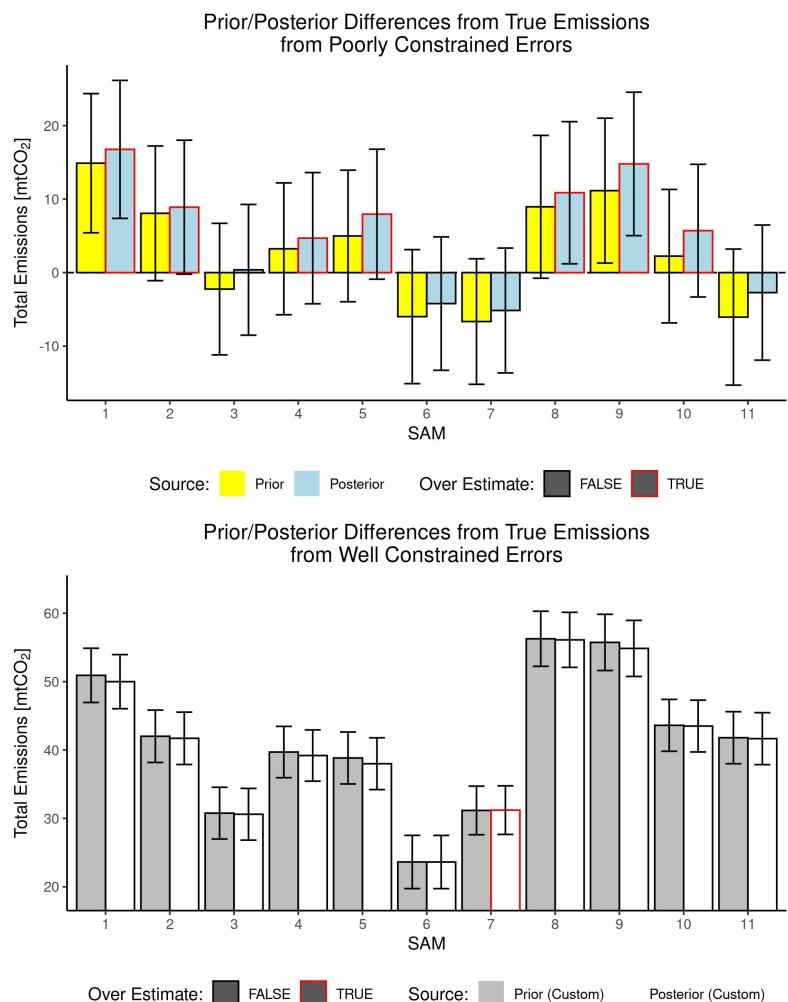

**Figure 16.** [Top] The hourly fluxes influencing each SAM were used to determine the total $CO_2$ emitted from the domain. Unmodified (poorly constrained) *prior* fluxes and their associated *posterior* estimates were compared against the "*true*" amount emitted. Cases where |*posterior - true*| is larger than |*prior - true*| are indicated with red borders. Error bars represent error derived from the *prior* and *posterior* covariance matrices. [Bottom] Same as above with customized (well constrained) flux errors.

from **Tab. 1** are first considered in **Fig. 18** where their percentage contributions to $\Delta XCO_2$ enhancements are shown. In the top set of panels, the average percentage contributions to $\Delta XCO_2$ from each emission category is spatially distributed for each SAM and averaged at a $0.1° \times 0.1°$ resolution. Furthermore, the per-sounding contributions from all SAMs are included in the accompanying box plots (bottom). A notable emission category is "Transportation" which had the most variation across soundings. This category contributed to 0% - 100% of the observed enhancements with the strongest contributions to the northwest of LA. More than half of the soundings over LA had a contribution from this category that was $\geq 50\%$.

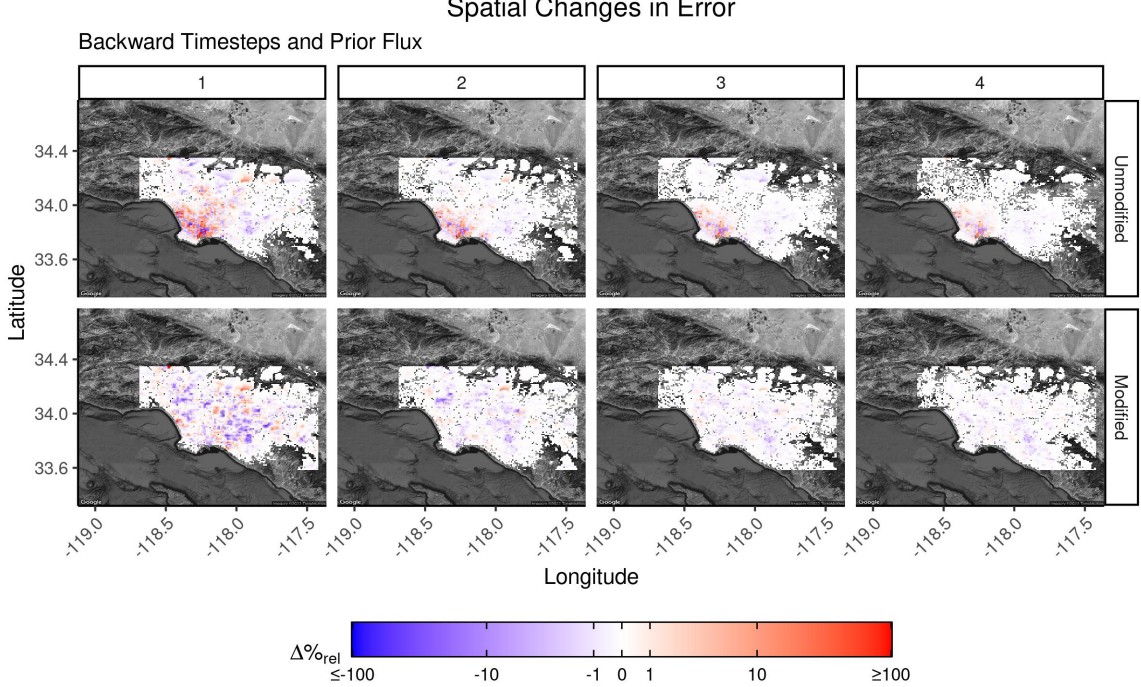

**Figure 17.** Presented here are the averaged changes in relative flux error provided by the inversion scheme. Backward timesteps associated with 11 SAMs were superimposed and averaged for the Los Angeles Basin. Columns indicate the first four hourly timesteps from the time of observation and rows indicate the type of *prior* emissions used. "Unmodified" refers to flux values that include ODIAC-VIIRS estimates while "Modified" refers to flux values that include area flux values from ODIAC-VIIRS and "*true*" LPS values from Vulcan 3.0. Small values in the range $\pm 0.01\%$ are excluded.

Following the Transportation category, contributions to enhancements from Manufacturing emissions also exhibited considerable variation; however, most of the large contributions were outliers. Enhancements were most influenced by this sector within LA and along the coast, as shown in the top panel of **Fig. 18**. Contributions weakened toward the northwest of the basin. Of the remaining two categories, the Power Industry exhibited small contributions across most soundings. The spatial distribution of these contributions revealed localized "hotspots" along the coast of LA. This emission category consisted of LPSs, which limited its influence on more distant enhancements. The "Buildings" category consisted of emissions from the commercial and residential sectors throughout LA. Almost all contributions from this sector were $< 50\%$ but spatially distributed, almost uniformly, across LA in $XCO_2$-space.

Results from the inversion scheme's ability to correct systematic biases in these emissions categories (described in **Sec. 2.5.4**) is presented in **Fig. 19**. Beginning with the first available SAM, **Eqn. 6** was applied to a *prior* emission inventory with flux values equal to 25% of Vulcan 3.0 fluxes ("*true*" emissions). Subsequent iterations increased the number of SAMs used by one, producing a new $\hat{\lambda}$ containing corrective values for each emission category. Since the initial bias in the *prior*

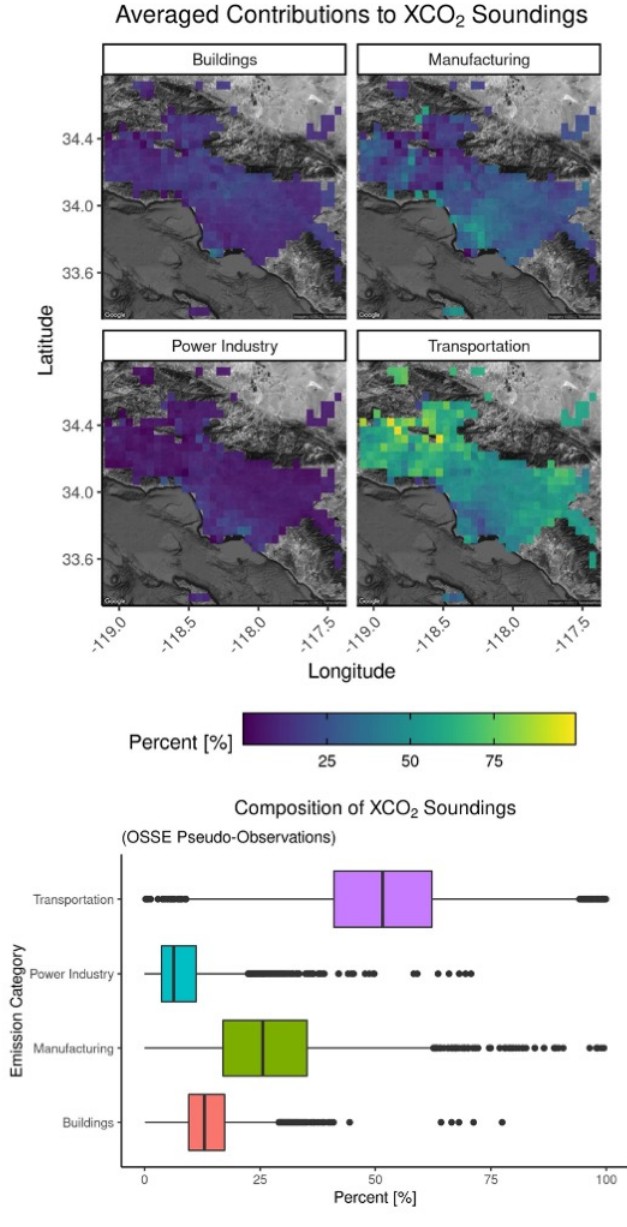

**Figure 18.** [Top] The percent contributions of emissions categories to the soundings in all available SAMs were superimposed and averaged at a $0.1° \times 0.1°$ resolution. [Bottom] The per-sounding contributions from each emissions category are represented as box plots.

emissions was known, a final scaling factor of 4 was desired for each emission category. The left panel of the figure shows two categories, Transportation and Manufacturing, quickly asymptoting to this value. Using only the first SAM, values in $\hat{\lambda}$ provide strong corrections for these two categories. The Power Industry is corrected slowly, likely due to its localized signal in





XCO$_2$-space, and never achieves full bias correction. The Building sector experiences only minor corrections, likely due to its small contribution to $\Delta$XCO$_2$ enhancements across LA (see **Fig. 18**).

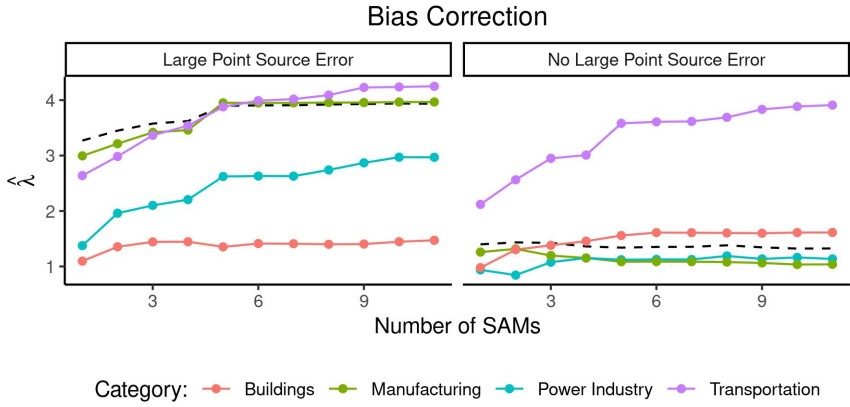

**Figure 19.** [Left] Results from a Bayesian inversion scheme set up to correct for systematic biases in *prior* emission inventories is presented here. The number of SAMs included in each iteration of the inversion is indicated by the x-axis and resulting corrective factors are shown on the y-axis. Errors associated with large point sources were included. [Right] Same as right with no large point source error included. The dashed black line shows the results from correcting the total flux rather than sectorally specific flux fields. In both scenarios, the prior is scaled to 25% of the "*true*" flux. Therefore, $\hat{\lambda} = 4$ would optimize the *prior* (excluding the Manufacturing and Power Industry categories when "no large point source error" is included).

     This scheme was applied to a second scenario where categories containing LPSs were not biased in the *prior* flux values. This setup was representative of areas where LPS locations and their associated emissions are well known but smaller area emissions are poorly constrained. As shown in the panel on the right, category emissions from the Transportation sector asymptote to

a corrective factor of 4; however, these iterations converge more slowly than the previous scenario. As expected, the Power Industry and Manufacturing categories maintain a corrective factor of 1 since no biases were included in these emissions. As in the previous scenario, the Buildings category is not well corrected.

## 4   Discussion

     As shown in **Fig. 6**, the daytime variation in the strength of $\Delta$XCO$_2$ pseudo-enhancements was statistically insignificant;
furthermore, the amount of soundings included in a SAM was not necessarily indicative of its information content. Since the primary driver of a SAM's information content is the magnitude of observed $\Delta$XCO$_2$, ensuring that the instrument's spatial coverage records the entire distribution of a city's enhancement should be prioritized. **Figure 1** presented several large SAMs made up of soundings to the northwest that are minimally influenced by surface fluxes. Future observations may be obtained over a smaller domain to increase the spatial density of strong XCO$_2$ signal relative to the number of observations. Additionally,
"tails" on the leading and trailing tracks of each SAM, outside the boundaries of the flux domain, are useful for calculating





background values (Wu et al., 2018). Initial differences between "observed" $\Delta XCO_2$ and corresponding *prior* estimates were key drivers of the Bayesian inversion scheme, as demonstrated in the first test case (**Sect. 3.1**). LPSs in the *prior* were not co-located with their "*true*" locations along the coast, producing significant corrections in the *posterior* $\Delta XCO_2$ estimates in this region. However, areas of smaller differences between *prior* and pseudo-enhancements to the east of the basin received

no correction. As originally demonstrated by Hogue et al. (2017), Gately, and Hutyra (2017), coarsening the resolution of the input grids reduced the magnitude of the mismatched cells' error associated with LPSs. In turn, this led to reduced corrections in *posterior* estimates of $\Delta XCO_2$ and increased biases of additional backwards timesteps

Although post-inversion differences between *posterior* estimates and "observed" $\Delta XCO_2$ were smaller than those of *prior* estimates, these corrections were provided at the cost of flux accuracy (Kaminski et al., 2001). Flux grid cells with large

uncertainty were corrected by the inversion, often by drastically increasing the *prior* flux value to account for the lack of a LPS; however, the magnitude of this increase was also applied to neighboring cells as prescribed by the spatial correlation among errors (see **Fig. 11**). This correlation "over-optimized" surrounding cells, raising their values far higher than necessary. This effect is most dominant during the optimization of the first backwards timestep and is localized to the southern portion of LA, where the largest LPS mismatches reside. Subsequent timesteps demonstrated the same feature but its magnitude decayed

quickly. The magnitude of increases and decreases in post-inversion errors were attenuated as input grids were coarsened, yet their effects lingered across more timesteps. Reinforcing similar findings by Kunik et al. (2019), these results demonstrated that the optimization of *prior* surface flux via space-based SAM observations is susceptible to poorly constrained LPS estimates and cannot be addressed by grid coarsening alone. The effects of spatial correlation among flux errors was also highlighted, suggesting that alternative correlation schemes may be useful. For example, spatial correlation at/near LPS locations could be

removed entirely, allowing LPS-specific cells to be corrected independent of their adjacent values.

Using surface observations, Oda et al. (2017) demonstrated that increasing the resolution of *prior* flux estimates led to better *posterior* constraints at the city-level; however, Yang et al. (2020) suggested that inversions are insensitive to input grid resolution when constraining city-level emissions with OCO-2 (column-averaged observations). Limited sensitivity to input grid resolution can be explained by the proximity of the measured $\Delta XCO_2$ to the urban area of interest. While most

surface-based measurements are obtained *within* urban areas, OCO-2 transects are often not direct overpasses but downwind of cities. Observing a signal "nearby" allows for more atmospheric mixing to occur before the signal is obtained, reducing the inversion's ability to resolve sub-city-level features; however, SAMs are collected directly over urban areas, minimizing the amount of mixing that occurs between the source and observed enhancement much like in-situ measurements.

Test Case 2 (**Sect. 3.2**) investigated post-inversion improvements from reduced atmospheric transport error. This exacerbated

the differences between post- and pre-inversion $\Delta XCO_2$ error. The inclusion of both a linear regression and a running average of the data (**Fig 12**) demonstrated that corrections driven by differences in *prior* and "observed" values were not linearly dependent. In the case of full transport error, soundings that initially differed by $\leq 0.5$ ppm did not demonstrate significant correction after the inversion was applied whereas soundings with larger initial differences exhibited the most correction. After the transport error was reduced, smaller *prior* enhancements demonstrated modest corrections while corrections provided by

larger enhancements were disproportionately larger. Although *posterior* $\Delta XCO_2$ estimates became closer to observed values





and the area of correction was larger after the transport error was reduced, biases introduced in the associated flux estimates were increased. This was evident in the hourly spatial distributions of error reduction presented in **Fig. 14**. As in the previous case, cells in the south of the basin with large *prior* flux error show improvement yet adversely influence neighboring cells. The magnitude and area of this feature are increased after the transport error is reduced. Similarly, a larger area in the east of

the basin demonstrates greater improvement in *posterior* flux estimates.

Observed $\Delta XCO_2$ enhancements are relatively small compared to errors associated with space-based observations. The instrument error used in this work is specific to the LA Basin and a more general error value could be $> 1$ ppm (Connor et al., 2016). Improvements in $\Delta XCO_2$ errors should be accompanied by reduced *prior* flux error to maintain balance between the error components of flux- and $XCO_2$-space in the cost function (**Eqn. 3**). The combination of transport error reduction

and corrections to the locations of LPSs (**Sec. 3.3**) reduced the errors in the south of the basin responsible for driving the optimizations in previous cases. As shown in **Fig. 17**, constraining the error from LPS locations within the *prior* dispersed the areas of increased bias in *posterior* outputs. Although small localized clusters of increased relative error exist, reductions are more prominent and total $CO_2$ values no longer over estimate the "*true*" totals; however, decreasing the error in flux-space also decreased the number of timesteps in which corrections were sustained. Although accurate spatial representation of LPSs

improved *posterior* estimates, Kunik et al. (2019) suggest temporal variation in LPS emissions is also a significant source of uncertainty.

In scenarios with reduced atmospheric transport error and "well behaved" flux errors, optimized results were restricted to the first few backwards timesteps; however, using SAMs collectively resulted in strong corrections to biased *prior* data. Roughly five SAMs were required to eliminate biases in the strongly contributing Transportation and Manufacturing sectors. Sectors

exhibiting weaker contributions to observed $\Delta XCO_2$ experienced less correction over time but additional SAM observations will further reduce this bias. The number of SAMs required to constrain sector-specific information from LA is similar to the number of OCO-2 transects required to constrain whole-city emissions (Ye et al., 2020). This inversion technique can be useful in regions that do not have ground-based $CO_2$ measurement networks or sparse/inaccurate data for emission inventories. Furthermore, the construction of emission inventories does not occur in real-time; therefore, the data products released often

cover previous years (Roten et al., 2022). Collections of dense $\Delta XCO_2$ observations from OCO-3 and other instruments taken throughout each year may be used to update data products from previous years.

This work used a Bayesian inverse method that was established in other studies (Kunik et al., 2019; Mallia et al., 2020; Turner et al., 2020). The approach used to characterize *prior* flux errors was used previously for the analysis of ground-based stationary and mobile $CO_2$ measurements. Novel methods of error characterization may be developed that are better suited

to the strengths of space-based observations; however, a detailed analysis of multiple error quantification methods is beyond the scope of this work. Given the interplay between spatial correlation and LPS uncertainty, the construction of a "dynamic" spatial length scale may provide a more robust spatial correlation matrix that can accommodate both LPS and area emissions. Additionally, assimilating surface-based measurements into the inversion scheme may offer further improvements in *posterior* estimates but the implementation of this hybrid approach would be limited due to lack of widespread ground-based observation





networks. Efforts to increase the utility of the bias correction scheme may recast this method to provide per grid cell corrections at a variety of temporal scales using collections of dense $XCO_2$ observations.

## 5 Conclusions

Observations are essential to understanding anthropogenic aspects of the carbon cycle. Space-based $XCO_2$ platforms are poised to provide dense observations of urban emissions at a near-global scale, enabling monitoring, reporting, and verification of
emissions-related policy at a variety of scales. To characterize the information made available by these instruments, an OSSE was developed which utilized a Bayesian inversion scheme to investigate the flux information contained in SAMs provided by the OCO-3 instrument. Results indicated that the optimization of *prior* emission estimates was susceptible to large flux errors driven by LPSs. Large initial error in $XCO_2$-space, dominated by transport errors, was also a contributing factor to the inversion's ability to provide corrections. Differences of $< 0.5$ ppm between *prior* and "*true*" $\Delta XCO_2$ demonstrated no
appreciable corrections in *posterior* estimates. Efforts to address the effects of large uncertainty included coarsening of the *prior* emissions inventory, reduction in transport error, and constraining the positions and uncertainties associated with LPSs. Results demonstrate that, under well constrained conditions, SAMs provide modest corrections for $\sim 1$ h prior to the time of $\Delta XCO_2$ observation; however, the use of multiple SAMs to correct systematic biases found in *prior* emissions estimates yielded promising results: large systematic biases were removed in the Transportation and Manufacturing sectors using only
five of the available SAMs while sectors with weaker fluxes were partially corrected. The methods and conclusions in this work provide a framework to investigate more robust methods of high-resolution Bayesian inversions of $XCO_2$. In the future, a dynamic approach to error/uncertainty quantification may adjust spatial sensitivities more appropriately, and corrective factors may be calculated at the cell-wise level rather than the sectoral level. As more observing platforms come online, these methods will become crucial in understanding $CO_2$ contributions from megacities around the world.



*Code and data availability.* The ODIAC emission data product is available from the Global Environmental Database hosted by the National Institute for Environmental Studies and can be downloaded at https://db.cger.nies.go.jp/dataset/ODIAC/. Vulcan 3.0 can be found at https://daac.ornl.gov/cgi-bin/dsviewer.pl?ds_id=1741. HRRR data can be obtained from the Air Resources Laboratory at https://www.ready.noaa.gov/READYmetdata.php. SMUrF data can be found at https://daac.ornl.gov/NACP/guides/Biogenic_CO2flux_SIF_SMUrF.html. The base inversion code used in this work can be found at https://doi.org/10.5281/zenodo.2655990. All maps were generated using the ggmap package (Kahle and Wickham, 2013).

*Author contributions.* D. Roten and J. Lin designed the test cases for this work. L. Kunik and D. Mallia contributed scripts from previous work, insights, and discussion about the methods used. D. Wu provided relevant results from the SMUrF model to approximate the error associated with the biosphere and T. Oda provided the ODIAC-VIIRS emission inventory. E. Kort provided suggestions and assisted to craft a narrative that aligns this work with the broader space-based $CO_2$ observation effort. All authors contributed to the editing of the final paper.

*Competing interests.* The authors of this work declare no competing interests.

*Acknowledgements.* The authors of this work wish to thank NOAA-ARL for maintaining a repository of HRRR meteorological files, the Center for High Performance Computing (CHPC) at the University of Utah for maintaining the computational resources necessary for the completion of this work, and Ben Fasoli for his dedication and assistance with matters of wrangling spatial data. The input from members of NASA's Orbiting Carbon Observatory mission during the early stages of this work has played a key role in the final product. This work was funded by the following NASA grants: 80NSSC19K0092, 80NSSC21K1065, 80NSSC21K1066, and 80NSSC21K1067.





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
