# Peer review of "The Information Content of Dense Carbon Dioxide Measurements from Space: A High-Resolution Inversion Approach with Synthetic Data from the OCO-3 Instrument"

_Atmospheric Chemistry and Physics, 2022_

## Referee Comment (RC2)

**Review of "The Information Content of Dense Carbon Dioxide Measurements from Space: A High-Resolution Inversion Approach with Synthetic Data from the OCO-3 Instrument" by Roten et al.**

This work applied observing system simulation experiments (OSSE) to evaluate the ability of OCO-3's SAM (synthetic) measurements to constrain fossil-fuel CO2 emissions using Bayesian-inversion modelling. The Vulcan 3.0 emission inventory and the X-STILT model were used to create synthetic CO2 enhancements. In the inverse model, Vulcan 3.0 emission inventory is considered as a true emission, whereas ODIAC emission inventory is considered as a prior emission.

This manuscript raises interesting and fundamental questions about the how the emission assessments are influenced by factors such as grid size of prior emission and transport error, as well as how will constraining in the locations and uncertainties of large point sources affect the inversion scheme and how aggregated SAMs affect bias correction. The answers to these questions could improve our understanding of inversion results as well as how to optimally use satellite measurements for emission assessments.

However, the manuscript is not easy to follow for people who are not inversion experts, and therefore requires some efforts for rewriting and restructuring. The results section can be more focused and shortened, so that the readers can easily grasp the outcome of the sensitivity studies. A table can be utilized to summarize the test conditions and achieved results (similar to Table 2). A better presentation will be needed for the readers to follow the concepts and appreciate the benefits of the results. The description of the Bayesian inversion scheme is a bit confusing in the end of section 2.4, i.e. how (6) is obtained. Also It is confusing that in the test 4 different criteria are used compared to other test cases.

Also I find it a confusing message that the posterior emission estimates are further from the truth compared to the prior emission, which are shown in Fig. 9, Fig. 13, and Fig.16. It introduces doubt to the general applicability/benefit of Bayesian inversion. In my opinion, the setup of the inversion framework regarding Q and R should be improved.

In addition, the authors uses terms such as "effectiveness of optimization", "corrective power" which sometimes just refer to the difference between posterior and prior estimates or the improvement of the fit to the data. The optimization of the emission estimates should be given by comparing with the true fluxes. Consequently, there are also misleading conclusions that may confuse the readers.

Apart from the general comments listed above, below please find some specific comments:

Line 114: Is the total emission of ODIAC and Vulcan different from one another? It's worth knowing because ODIAC typically underestimates road emissions because it uses night-time light data as a proxy.

Line 164: The first term on the right hand side...

Line 170: "R reflects uncertainties in DXCO2 observations from various components" is a bit misleading because it sounds like it only consists of observation errors. R consists also of transport errors, etc. as listed in Table 3.

Line 180: and instrument error ($\varepsilon$). Could you please elaborate how did you incorporate other error sources such as the transport error?

Figure 3: the color bar should be denoted as DXCO2 [ppm]

Figure 5: Histogram plot of difference between prior (customized and non-customized) and true emission will be more intuitive.

Line 328: "It demonstrates that the effectiveness of the optimization is directly proportional to the observed enhancement"
This, to our understanding, implies that good emission optimization necessitates a significant increase in CO2. In most low wind speed cases, CO2 will be significantly increased, but transport error could be high. Therefore, it is preferable to see an error bar in plot 6 (c). In addition, I do not think the effectiveness of optimization can be represented by Posterior flux – Prior Flux, it should be compared with the "true flux".

Figure 6 (b-d) and section 3.1.1 (319-334): Similar to the point before, to understand the inversion's ability to optimize estimates, the posterior flux should be compared with true flux (posterior flux - true flux) i.e., corrective power. If so, the difference between posterior flux and prior flux (posterior flux – prior flux), i.e., amount of correction, don't indicate the inversion's ability. If my understanding is correct, then it is also applicable to figure 7, 8, 12 and 15, and its discussion part.

Figure 9: The y axis should be "differences" in total emissions. It also applies to Figure 13 and 16. I would also write "Overestimate" instead of "Over Estimate"

Line 519: The background approach mainly accounts for major uncertainty. In this study, the authors used synthetic CO2 enhancement. So, they cannot assess the inversion ability for varying background uncertainty. However, in the paper, they only considered the background error from one paper (Kiel et al, 2021). The authors could consider and discuss the background approaches from other cases. For example:
Wu, D., Liu, J., Wennberg, P. O., Palmer, P. I., Nelson, R. R., Kiel, M., and Eldering, A.: Towards sector-based attribution using intra-city variations in satellite-based emission ratios between CO2 and CO, Atmos. Chem. Phys. Discuss. [preprint], https://doi.org/10.5194/acp-2021-1029, in review, 2022

---

## Author Comment (AC1)

We would like to thank the editor for considering this manuscript for publication and the reviewers who offered many useful suggestions for improvements. The incorporation of these suggestions led to an extensive rewrite which better addresses modeling errors, presents the inversion scheme more clearly, and removes figures containing redundant information. We now find the manuscript and its updated results provide more focused conclusions.

Below, you will find the original reviewers' comments in bold with our responses in red.

**Reviewer #1**

**The manuscript "The Information Content of Dense Carbon Dioxide Measurements from Space: A High-Resolution Inversion Approach with Synthetic Data from the OCO-3 Instrument" by Roten et al. assesses the potential of XCO2 "Snapshot Area Maps" (SAMs) from OCO-3 for the quantification of the CO2 emissions from large cities based on a classical Bayesian atmospheric inversion framework. It relies on tests with pseudo XCO2 data over Los Angeles.**

**The inversions of the anthropogenic CO2 emissions from cities based on satellite data receive a growing interest with the analysis of OCO-2 and OCO-3 city plume transects or images, the preparation of new generations of satellite XCO2 imagers, and the development of dedicated inversion systems. A wide range of studies have been published on this topic using both OCO-2/3 or synthetic XCO2 data. The authors of this manuscript have developed material (tools, simulations, experimental protocols, diagnostics) that can support the derivation of new insights and learnings in this field of activity. Their experiments bring some interesting results.**

**However, 1) the manuscript requires a major general rewriting and 2) I have concerns regarding the specific configuration of the experiments or regarding the conclusions raised from the results.**

**(1) The reading of the manuscript is laborious because of inappropriate or vague wordings and notations, and because of a lack of rigor and precision. Efforts, reasoning and some good knowledge of atmospheric inversion are often needed to unravel the meaning of the text.**
We appreciate this comment and have revised the manuscript throughout, including modified wording to enhance the clarity for readers that may not have a background in inversions. We have increased the detail in Section 2.4 to further explain some of the components of the process. Furthermore, the Results section was condensed, removing all non-essential and/or redundant plots. This allowed us to further focus our discussion and make the manuscript more readable. We have sought to address the "inappropriate or vague wordings/notations" specified by the Reviewer in detail below.

**(2) The abstract and introduction provide many meaningless and random statements. The abstract is hardly informative because its statements lack context. The introduction weaves between general considerations on the CO2 atmospheric inversions and indications that correspond to city scale applications only. It is sometimes difficult to connect a statement to the corresponding reference to a past publication. The discussions on the ground based**

**networks and on the "increased spatiotemporal" coverage of OCO-3 compared to OCO-2 are a bit misleading. The justification for the use of pseudo data experiments ("the use of synthetic data from OCO-3 eliminated the potential for systematic biases from local CO2 emissions reductions during the COVID-19 pandemic lockdowns and biases in preliminary data from OCO-3") is a bit puzzling and highlights the need for a clearer rationale for the specific analysis conducted in this new study.**

We edited the abstract and included some additional statements that highlight the overall goal of this work (Lines 5 - 6 of the Abstract). The Introduction of this manuscript was heavily edited to be more cohesive and provide stronger motivation for our study. We focused on the appropriate wording of our examples to better illustrate how in-situ inversions have laid the framework in the space-based inversion community while highlighting the utility of our current work. We also added a statement about using an OSSE approach so the "true" emissions can be known (Line 74). This makes the rationale of using synthetic data more clear.

In regards to the misleading OCO-2/OCO-3 statement, we found that the inclusion of the word "multiple" may lead the reader to think that SAMs were acquired multiple times a day for the same target location. The word "multiple" has been removed and the statement, in the context of the section is clearer. (Lines 66 - 69 of the Introduction)

**(3) The recollection of the critical assumptions and parameters of the modeling and inversion configurations (e.g. regarding the set-up of the control vector as a function of the test cases, or regarding the precise set-up and iterative process of test case 4) from section 2 is laborious. The end of section 2.4 is particularly confusing. The information is not properly organized. The presentation of the diagnostics in section 3 lacks of clarity and bears many little missteps. The title of the manuscript itself could be rethought to be more informative about the purpose of the study (e.g. about the focus on the monitoring of anthropogenic CO2 emissions from cities).**

These sections have been extensively edited. We feel that the condensed version of Section 3 will be more readable. Redundant plots were removed from the main text and replaced with summaries in the text while test case parameters are reiterated at the beginning of each subsection of the results. Our intention is that this new iteration of the results section will address the same findings as our first draft but in a more digestible way.

**(4) A major result from the experiments is the increase of errors in the estimate of the emissions from the inversion in the tests cases 1 and 2 (not only at "sub city level" as suggested by the abstract, but also at the city level). These test cases are those for which the differences between the true and prior estimate of the emissions are appropriately characterized by the difference between two inventories that are widely used in the community. The analysis reveals that the explanation for such an increase of errors is due to the poor adequacy between the spatial correlations in the Q matrix and the actual discrepancies between the two inventories. Indeed, when considering anthropogenic emissions within a city, exponentially decaying spatial correlations (inherited from large scale inversion practices) hardly make sense and the results confirm it. Such correlations may make more sense if splitting the control vector between the different sectors of activities with no correlation of uncertainties across the sectors. But even in that case, the size of cities and the dynamics of the emissions hardly justifies correlations of uncertainties decreasing in space (in an isotropic way). More details about the diagnostics of the**

correlation lengths may feed this discussion, but this computation seems to be based on a single occurrence of error map (lines 300-305 are not really clear) which can be misleading here: the significant correlations in space probably arise from the areas of the map where the different sectors are relatively well mixed and the test does not account for the fact that the correlation rises up again further between areas that bear similar emission sectors (?). As briefly envisaged in section 4, rather than building an hybrid ("custom") prior estimate of the emissions to overcome the problem, I believe that the authors should have improved the set-up of the control vector and of the corresponding matrix Q, especially since results from test case 4 reveal that the use of this custom prior does not really solve for the lack of improvement in the emission estimates. The implicit conclusion suggested by the manuscript that the large point sources within the city should be correctly geolocated to get good estimates of the city total emissions cannot rely on the current set of results.

In the introduction, we point out that this work uses an inversion approach that has already been established (Line 78). We follow the construction methodology of the **Q** matrix as prescribed in Kunik et al., (2019) who also used this method for urban fluxes. We added additional citations where similar iterations of this method were used (Lauvaux et al., 2016 10.1002/2015JD024473; and Nevison et al., 2021 10.1002/2017GB005759). Both Vulcan 3.0 and ODIAC-VIIRS are 1km resolution inventories that cover CONUS. ODIAC-VIIRS does not provide sector-specific emissions data. Therefore the tradeoff of using high-resolution prior information is that only the differences in total fluxes can be used to generate the **Q** matrix. We use our results to address this issue and show that more robust methods are needed to quantify fluxes for space-based applications (the signal-to-noise ratio is small). This is addressed in the Discussion section (Lines 547 - 564).

We would also like to clarify points relating to the final statement in this reviewer comment. Our work demonstrates that cells with large uncertainties relative to their prior emission values can greatly influence the estimated flux in that area. If the uncertainty in the geolocation of large point sources such as power plants is incorporated into the construction of spatially explicit emission inventories, there is a chance of producing single cells with large uncertainty (Hogue et al., 2016 & 2017). Our goal was not to suggest that correctly located large point sources leads to good estimates of city $CO_2$. We demonstrate that correctly located point sources (and thus lower cellwise uncertainty) *improve* the sub-city-level results of flux estimates, not necessarily "get good estimates". (Hogue et al., 2016 10.1002/2015EF000343 & Hogue et al., 2019 10.1007/s11027-017-9770-z )

**(5) My understanding is that the perturbations applied to the "true" XCO2 field in order to generate pseudo data are not consistent with the set-up of the R matrix in the inversion system. This could provide insights on the skill of the inversion when the inversion configuration does not properly characterize the statistics of the actual errors in the model vs. data misfits. But this needs to be properly handled, analyzed and discussed. Here, the manuscript ignores this lack of consistency and raises conclusions that can be highly misleading, in particular regarding the impact of "decreasing the model error" (actually, of decreasing R but the "true" model errors are null in these experiments). If there is no transport model errors in the model vs. data misfits, then decreasing or increasing R simply leads to fitting more or less the data. If the data drive the inversion in the wrong direction (which is the case in Test case 1), decreasing R will increase the problem (which is the case in test case 2). That does not easily say something about what would happen if there would**

**be small or large transport errors in the model vs. data misfits. This point connects to the previous one and the lack of consistency between the assumption by the inversion system that prior uncertainties follow the distribution N(0,Q) and the actual differences between ODIAC and Vulcan.**

Thank you for pointing out this oversight. We have since incorporated these errors into our inversion scheme (Eqn. 6), making **R** consistent with the expectations of this comment. To help readers who are unfamiliar with Bayesian inversions, the "Sources of Error" section was moved to immediately follow the development of the inversion (Sect. 2.4). A new term, $\Delta\varepsilon$, was added to the presentation of the $\hat{s}$ equation to represent the inclusion of all errors from **Tab. 2**. The text in these sections was updated accordingly. We have emphasized that a "reduction in transport error" is merely a reduction in the values of the **R** matrix and could be achieved by a reduction in the uncertainty of an alternative parameter.

**(6) We hardly understand why the analysis of the average emission estimates based on the joint use of multiple SAMs in Test case 4 relies on an experimental set up which is completely different from the other ones. Why considering a huge bias (a factor 4, which hardly applies to city scale inventories ?) and no other errors in the prior estimate of the emissions here? The results from this test case are difficult to connect to the others and the huge error on the prior estimate of the emissions prevents this experiment from convincing us about the potential of the SAMs. In a general way, the analysis hardly provides quantitative analysis of the typical precision of the emission estimates from the inversions, which is a key index of this potential.**

A distinction between Test Cases 1-3 and Test Case 4 is the use of multiple SAMs (TC 4) vs. individual SAMs (TC 1-3). The first three test cases are intended to quantify the information content of individual SAMs. Similar work was conducted by Ye et al. (2020) where optimized scaling factors were calculated for individual OCO-2 transects and considerable variation among scaling factors was demonstrated. In an effort to increase the constraints in the inversion, multiple SAMs were used in Test Case #4.

It is unclear from this comment whether the reviewer is referring to **Q** or **K** as the "prior estimate". Since TC 4 does not rely on prior *flux* errors (**Q**), we assume that **K** was intended. To address this issue, the latest iteration of the experiment incorporated all errors mentioned in Sect. 2.5 into the **R** matrix. Specifically, the new **R** matrices generated to address this issue in TC 1-3 were used in TC 4 (Creating a block matrix of **R**'s). We also included an updated figure which presents the error associated with each step of the iterative process. The calculation for this error was added and discussed in Sect. 2.5.

Regarding the size of the introduced bias: many of the developing countries that are signatories of the Paris Agreement (2015) are committed to producing emission inventories. In places where the infrastructure to do this is incomplete, considerable errors and biases could exist. In this work, Test Case 4 considers a situation where a considerable bias exists due to inadequate accounting methodology. We cited Gurney et al., 2021 to highlight this possibility (10.1038/s41467-020-20871-0). Results indicate that at least two sectors are partially corrected, with the transportation sector responding considerably well. However, after some discussion, we agree with the Reviewer that reducing the prior to 0.25 of the true emissions was indeed too large of a bias. We have since changed this bias to a reduction of 0.5 with results behaving similarly to previous outcomes.

**(7) Los Angeles appears to be a very complex case for city scale inversions due to the surrounding topography and ocean. The modeling of the CO2 transport over such a city is challenging (from this point of view, lines 214-217 can be misleading). By using the same transport model to simulate the pseudo data and for the inversion (i.e. by using a perfect transport model for the inversion), the study avoids this issue. This should be properly discussed.**

This was addressed during the rewriting of **Sects. 2.1** and **2.4** (and briefly in the Introduction (Lines 73-74)). We demonstrate the interest of the area by pointing out several other studies that focus on the LA Basin. This domain is frequently studied and the results from our current work will help inform future space-based studies of LA.

Using the atmospheric transport from X-STILT as "truth" is acknowledged in Lines 179-182 and by the inclusion of *H in **Eqn. 5**. This emphasizes that the forward model is being treated as "truth" while the appropriate errors (including the transport errors) are included in $\Delta\varepsilon$.

**(8) The characterization of the "background" field underlying the CO2 emitted by the city appears to be a critical source of uncertainty for city scale inversions (in general, it does not seem to be as easy as suggested by lines 271-272). The size of OCO-3 SAMs may actually be limited for the characterization of the background of cities such as Los Angeles. This should also be appropriately taken into account when discussing the optimal spatial sampling in section 4.**

In hindsight, we found our discussion of spatial sampling could be misleading. We now discuss the importance (and limitations) of calculating an appropriate background value and how the use of an OSSE makes simplified assumptions about this component (i.e. - the background is subtracted out "perfectly"). We have also cited an additional paper that makes use of a similar background calculation method (Wu et al., 2022) in the Methods section and pointed out the importance of large spatial coverage in the first paragraph of the Discussion section. Any further discussion of SAMs' limitations will best be addressed in future work.

**(9) Despite ignoring these two critical sources of errors (in addition to the "bias" in the real retrievals of XCO2 data, such as those mentioned at line 77), the inversion hardly provides convincing results for the estimate of the city total emissions (improvements at city scale in test case 3 are nearly negligible, and see my concerns regarding test case 4). Opposed to the last statement of the abstract (and to those of the final lines of the introduction), it does not really demonstrate the need for such measurements.**

We agree that applying the inversion to *individual* SAMs produces very small optimizations in surface flux. This is a key conclusion of the paper. However, we would like to point out that the statement referenced in the abstract is specifically about the use of *multiple* SAMs (as written: "The aggregation of multiple SAMs prove to be effective in reducing systematic errors…"). Even with a reduction in prior bias used in Test Case #4 (from a bias of 0.25 of *true* emissions to 0.5 of *true* emissions; see Reviewer #1, Comment #6), all emission sectors were at least partially corrected. As efforts to reduce uncertainty in myriad components of this inversion process move forward, frequent XCO2 observations over a long period of time will assist in (at least partially) constraining emissions from sector-specific sources at a global scale, providing a guide to inventory builders to address biases and shortcomings in accounting methods. In regards to the inadequacies of the **R** matrix, we have since rewritten Sect. 2.4 (Bayesian Inversion) and

incorporated more robust values as suggested by the reviewer. (See Reviewer #1, Comments #5 and #7)

**(10) A point regarding the set-up of R: first, lines 267-268 are misleading. Prior XCO2 vs. data misfits in the XCO2-space include the transport of the errors in the prior estimate of the fluxes. Then, the misfits between the pseudo data and the prior XCO2 concentration seems not to include transport model, background and biosphere errors. Finally, by construction, there is no spatial correlation between the instrumental errors which have been used to perturb the pseudo-data. Therefore, it seems that the derivation of the spatial correlations for R based on the comparison between the pseudo-data and the prior XCO2 (l. 286) does not make sense.**
Again, we appreciate the Reviewer for bringing this to our attention. The original construction of our **R** matrix was indeed inadequate. We have since rectified this problem and addressed it within the text and subsequent analyses. You will find that our **R** matrix is now inclusive of all errors discussed in the manuscript.

**(11) l340-343: I do not agree, the level of improvement of the fit to the data is not an index of the potential to reduce the error in the emission estimates, or at least not at scales larger than the resolution of the control vector (especially when the R and B matrices are not consistent with the actual errors as here). Controlling emission at higher spatial resolution provides more degrees of freedom to fit the data along with capabilities to decrease errors in emission estimates at this higher resolution. But one cannot say much more about this with the diagnostics provided there ? This concern propagates to lines 347-348 and subsequent considerations until the end of section 3.1.1.**
This figure (and similar figures from the following subsections) and its accompanying interpretation has been removed from the revised manuscript. We found that it detracted from the main points of the paper.

**(12) Regarding the discussion at lines 541-548: what does Figure 9 say about it (either the posterior estimates of the emissions or the uncertainties in these estimates)?**
Thank you for pointing out this missed opportunity! While **Fig. 9** was removed from the results section and replaced with a text summary, these results were still incorporated into the discussion.

**Reviewer #2**

**This work applied observing system simulation experiments (OSSE) to evaluate the ability of OCO-3's SAM (synthetic) measurements to constrain fossil-fuel CO2 emissions using Bayesian-inversion modelling. The Vulcan 3.0 emission inventory and the X-STILT model were used to create synthetic CO2 enhancements. In the inverse model, Vulcan 3.0 emission inventory is considered as a true emission, whereas ODIAC emission inventory is considered as a prior emission.**

**This manuscript raises interesting and fundamental questions about the how the emission assessments are influenced by factors such as grid size of prior emission and transport**

error, as well as how will constraining in the locations and uncertainties of large point sources affect the inversion scheme and how aggregated SAMs affect bias correction. The answers to these questions could improve our understanding of inversion results as well as how to optimally use satellite measurements for emission assessments.

**(1) However, the manuscript is not easy to follow for people who are not inversion experts, and therefore requires some efforts for rewriting and restructuring. The results section can be more focused and shortened, so that the readers can easily grasp the outcome of the sensitivity studies. A table can be utilized to summarize the test conditions and achieved results (similar to Table 2). A better presentation will be needed for the readers to follow the concepts and appreciate the benefits of the results. The description of the Bayesian inversion scheme is a bit confusing in the end of section 2.4, i.e. how (6) is obtained. Also It is confusing that in the test 4 different criteria are used compared to other test cases.**

We edited the Introduction of this manuscript to better "flow" between examples of inversions, better introducing the need for space-based inversion schemes (See Reviewer #1, Comments #1 and #2). Furthermore, to make the inversion scheme clear, **Sect. 2.4** was rewritten to better address the setup and formulation of **Eqn. 5**. Furthermore, this equation was rewritten to more explicitly demonstrate its components (i.e. - the inclusion of the *H and $\Delta \varepsilon$ terms.) As suggested, the results section was shortened by removing figures that were not crucial to the *main* points of the manuscript. Several figures presented redundant information that could be gleaned from other sources within the manuscript. When possible, the main points of figures were summarized in the text while the figure itself was removed. Additionally, we have provided more detail about the origin of **Eqn. 6**.

**(2) Also I find it a confusing message that the posterior emission estimates are further from the truth compared to the prior emission, which are shown in Fig. 9, Fig. 13, and Fig.16. It introduces doubt to the general applicability/benefit of Bayesian inversion. In my opinion, the setup of the inversion framework regarding Q and R should be improved.**

These figures have been removed from the revised draft; however, their results were summarized within the text. This work applied a previously-established method (Kunik et al., 2019) and uses the prescribed construction of the **Q** matrix. Developing and testing new methods of constructing **Q** and **R** is left as a future endeavor and possible approaches are discussed. This is addressed in the Introduction (Lines 78 - 80), Section 2.5.2 (Line 231), and Discussion sections of the manuscript. (See response to Reviewer #1, Comment #4.)

**(3) In addition, the authors use terms such as "effectiveness of optimization", "corrective power" which sometimes just refer to the difference between posterior and prior estimates or the improvement of the fit to the data. The optimization of the emission estimates should be given by comparing with the true fluxes. Consequently, there are also misleading conclusions that may confuse the readers.**

Error reductions in XCO2-space are no longer calculated by comparisons to observations. Instead, posterior estimates are compared to the true emissions. Many of the figures bearing these terms in their titles were removed to shorten the results section (while preserving their "messages" within the text). Often, key features from one figure could be gleaned from another. Throughout the revising process, vague terminology was removed and/or reworded to be more consistent with typical terminology.

**(4) Line 114: Is the total emission of ODIAC and Vulcan different from one another? It's worth knowing because ODIAC typically underestimates road emissions because it uses night-time light data as a proxy.**

Yes. ODIAC does underestimate road emissions as shown in **Fig. 1**. This feature of Vulcan 3.0 is also highlighted in Lines 113 - 116. Additionally the difference between ODIAC-VIIRS and Vulcan 3.0 can be found in Lines 361-365. (On average, ODIAC-VIIRS estimates ~3 mtCO2 larger than Vulcan 3.0.)

**(5) Line 164: The first term on the right hand side…**

Thank you for the suggestion! This statement has been included in the revised manuscript.

**(6) Line 170: "R reflects uncertainties in DXCO2 observations from various components" is a bit misleading because it sounds like it only consists of observation errors. R consists also of transport errors, etc. as listed in Table 3.**

Our initial approach in this work was, in fact, inadequate. We have since added these errors to the R matrix and discussed them at length in the methodology section. This section was restructured to move the discussion of error sources closer to the introduction of the inversion scheme. The major points of the paper have remained largely unchanged. (For more comments from the authors, refer to Review #1, Comments #5 and #10.)

**(7) Line 180: and instrument error (ε). Could you please elaborate how did you incorporate other error sources such as the transport error?**

Yes. This discussion has been added in sections 2.4 and 2.5 of the revised manuscript. (Also, see our response to the preceding comment.)

**(8) Figure 3: the color bar should be denoted as DXCO2 [ppm]**

Agreed. Thank you for pointing out this oversight.

**(9) Figure 5: Histogram plot of difference between prior (customized and non-customized) and true emission will be more intuitive.**

We considered a histogram plot of these values; however, the process of constructing the customized inventory only removes the largest differences (+/- 1000 umol/m2/s). Only the fringes of the distributions in the histograms changed while the bulk of values remained the same. This slight change was hard to see in the form of a histogram. So, we have regenerated figure 5 with a zoomed in map. This allows readers to better visualize the removal of the large point sources.

**(10) Line 328: "It demonstrates that the effectiveness of the optimization is directly proportional to the observed enhancement" This, to our understanding, implies that good emission optimization necessitates a significant increase in CO2. In most low wind speed cases, CO2 will be significantly increased, but transport error could be high. Therefore, it is preferable to see an error bar in plot 6 (c). In addition, I do not think the effectiveness of optimization can be represented by Posterior flux – Prior Flux, it should be compared with the "true flux".**

After consideration, we have determined that the results from this figure detract from the main messages of the manuscript. In an effort to shorten the document and focus on other findings, these results have been removed and will be revisited in later works.

**(11) Figure 6 (b-d) and section 3.1.1 (319-334): Similar to the point before, to understand the inversion's ability to optimize estimates, the posterior flux should be compared with true flux (posterior flux - true flux) i.e., corrective power. If so, the difference between posterior flux and prior flux (posterior flux – prior flux), i.e., amount of correction, don't indicate the inversion's ability. If my understanding is correct, then it is also applicable to figure 7, 8, 12 and 15, and its discussion part.**

Agreed. The calculations have been changed to reflect differences between the prior and true XCO2.

**(12) Figure 9: The y axis should be "differences" in total emissions. It also applies to Figure 13 and 16. I would also write "Overestimate" instead of "Over Estimate"**

These plots have been removed and their findings are now summarized within the text.

**(13) Line 519: The background approach mainly accounts for major uncertainty. In this study, the authors used synthetic CO2 enhancement. So, they cannot assess the inversion ability for varying background uncertainty. However, in the paper, they only considered the background error from one paper (Kiel et al, 2021). The authors could consider and discuss the background approaches from other cases. For example: Wu, D., Liu, J., Wennberg, P. O., Palmer, P. I., Nelson, R. R., Kiel, M., and Eldering, A.: Towards sector-based attribution using intra-city variations in satellite-based emission ratios between CO2 and CO, Atmos. Chem. Phys. Discuss. [preprint], https://doi.org/10.5194/acp2021-1029, in review, 202**

In hindsight, we realize how this statement may be misleading. In the revised version of the draft, this statement has been removed. We have added a few statements about calculating the background XCO2 value in the Discussion section.